


**Precipitation of dolomite from seawater on a Carnian coastal plain (Dolomites, northern**
**Italy): evidence from carbonate petrography and Sr-isotopes**
Maximilian Rieder[1], Wencke Wegner[2], Monika Horschinegg[2], Stephanie Klackl[1], Nereo
Preto[3], Anna Breda[3], Susanne Gier[1], Urs Klötzli[2], Stefano M. Bernasconi[4], Gernot Arp[5],
Patrick Meister[1]
[1] Department of Geodynamics and Sedimentology, University of Vienna, Althanstr. 14, 1090 Vienna, Austria
[2] Department of Lithospheric Research, University of Vienna, Althanstr. 14, 1090 Vienna, Austria
[3] Department of Geosciences, University of Padova, Via Gradenigo 6, 35131 Padova, Italy
[4] Geological Institute, ETH Zürich, Sonneggstr. 5, 8092 Zürich, Switzerland
[5] Geoscience Centre, University of Göttingen, Goldschmidtstr. 3, 37077 Göttingen, Germany
Correspondence: Patrick Meister (patrick.meister@univie.ac.at)
**Abstract.** The geochemical conditions conducive to dolomite formation in shallow evaporitic
environments along the Triassic Tethyan margin are still poorly understood. Most of the
Triassic dolomites in the Austroalpine and the South Alpine realm are affected by late
diagenetic or hydrothermal overprinting, but recent studies from the Carnian Travenanzes
Formation (South Alpine) provide evidence of primary dolomite. Here a petrographic and
geochemical study of the dolomites intercalated in a 100-m-thick Carnian sequence of distal
alluvial plain deposits is presented to gain better insight into the conditions and processes of
dolomite formation. The dolomites occur as 10- to 50-cm-thick homogenous beds, mm-scale
laminated beds and nodules associated with palaeosols. The dolomite is nearly stoichiometric
with slightly attenuated c-reflections. Sedimentary structures indicate that the initial primary
dolomite or precursor phase consisted largely of unlithified mud. Strontium isotope ratios
($^{87}Sr/^{86}Sr$) of homogeneous and laminated dolomites reflect Triassic seawater, suggesting
precipitation in evaporating seawater in a coastal ephemeral lake or sabkha system. However,
the setting differed from modern sabkha or coastal ephemeral lake systems by seasonally wet
conditions with a significant siliciclatic input and inhibition of significant lateral groundwater



flow through impermeable clay deposits, thus representing a non-actualistic system in which
dolomite formed along the ancient Tethyan margin.
**Keywords** Dolomite, Sr-isotopes, sabkha, alluvial plain, peritidal platform, Travenanzes
Formation, ephemeral lake, authigenic carbonate.
**1 Introduction**
The formation of dolomite $[CaMg(CO_3)_2]$ under Earth surface conditions in modern and
ancient environments is still a major unsolved problem in sedimentary geology. Dolomite
does not precipitate from modern open ocean water, apparently, because its nucleation and
growth is inhibited by a high kinetic barrier. For the same reason its precipitation under
laboratory conditions has been difficult (cf. Land, 1998), and therefore the factors that may
have influenced dolomite formation through Earth history, giving rise to a significant part of
the sedimentary record, also remain poorly constrained. Van Tuyl (1916) discussed several
competing theories, one of which was the chemical theory, where dolomite is a primary
precipitate, hence, forming as a result of the conditions prevailing in the depositional
environment. In contrast, stable isotope and fluid inclusion data often indicate that massive
dolomites formed due to replacement of precursor calcium carbonate during burial diagenesis,
i.e., at higher temperature and under conditions decoupled from the ancient depositional
environment. Chilingar (1965) suggested that the portion of dolomite in carbonates increases
with geological age, implying a replacement during burial. However, burial dolomitization
requires a mechanism pumping large volumes of Mg-rich water through porous rock (Machel,
2004) and is not always a viable process. There is evidence that at certain times in Earth's
history, large amounts of dolomite could have formed under near-surface conditions
(penecontemporaneous dolomite), and several studies linked the abundance of dolomite to



secular variations in seawater chemistry, with preferred dolomite formation during times of
"calcite seas" (Given and Wilkinson, 1987; Warren, 2000; Burns et al., 2000).
In the Tethyan realm, penecontemporaneous dolomite formation seems to have prevailed
during the Triassic (Meister et al., 2013, and references therein), in an "aragonite sea", while
elsewhere dolomite was not particularly abundant (cf. Given and Wilkinson, 1987). In Norian
shallow water dolomites of the Dolomia Principale, Iannace and Frisia (1994) measured
oxygen isotope values as positive as +3.5‰, suggesting a formation at Earth surface
temperatures, whereas dolomites of the overlying early Jurassic units typically show
signatures of burial diagenetic overprint. Frisia et al. (1994) interpreted these dolomites to be
an early diagenetic replacement of precursor carbonate. In a recent study, Preto et al. (2015)
suggested that the dolomites of the Carnian Travenanzes Formation (Fm.) in the Venetian
Alps are primary precipitates, i.e. they precipitated directly from a solution in the sedimentary
environment and not by replacement of a precursor phase during burial. This interpretation is
based on high-resolution transmission electron microscope (HR-TEM) analysis showing
nanometre-sized crystal aggregates within single micron-scale dolomite crystals. The nano-
crystal structures were not replaced by any of the dolomite phases described by Frisia and
Wenk (1993) in Late Triassic dolomites of the Southern Alps, and they show similarity to
dislocation-ridden Mg-rich phases observed in modern sabkha dolomite and interpreted as
primary (Frisia and Wenk, 1993). This finding is intriguing, not only because it is consistent
with primary dolomite formation already discussed by Van Tuyl (1916) and observed in many
modern environments (e.g., Sabkha of Abu Dhabi: Illing, 1965; Wenk et al., 1993; unlithified
dolomite is also mentioned in Bontognali et al., 2010; and Court et al., 2017; Deep Springs
Lake, California: Jones, 1965; Clayton et al., 1968; Meister et al., 2011; Coorong Lakes: Von
der Borch, 1976, Rosen et al., 1989, Warren et al., 1990; Brejo do Espinho, Brazil; Sánchez-
Román et al., 2009; Lake Acigöl, Turkey: Balci et al., 2016; Lake Neusiedl, Austria:
Neuhuber et al., 2015; Lake Van: McCormack et al., 2018), but it also provides a window into



ancient primary dolomite formation pathways. This finding is also consistent with recent
experiments by Rodriguez-Blanco et al. (2015), demonstrating a nano-crystalline pathway of
dolomite nucleation and growth. Critically, nanometre size nuclei show a different surface
energy landscape compared to macroscopic crystals, allowing for potentially lower energy
barriers, perhaps modified by organic matter, microbial effects, clay minerals or particular
water chemistry, and thus, promoting a spontaneous precipitation of dolomite.

The interpretation of primary dolomite in the Travenanzes Fm. needs further validation by

nano- and atomic scale analyses and further petrographic and geochemical investigations to
establish the environmental and geochemical conditions on this Carnian platform. In
particular, the origin of ionic solutions conducive to dolomite formation is still unclear.
Comparison with modern environments shows that ionic solutions may either be seawater-
derived, as shown for the sabkhas along the Persian Gulf coast, where several hydrological
mechanisms were discussed (Adams and Rhodes, 1960; Hsü and Siegenthaler, 1969;
McKenzie et al., 1980, McKenzie, 1981; see Machel, 2004, for an overview; cf. also Teal et
al., 2000), or derived from continental groundwater, as shown for the coastal ephemeral lakes
of the Coorong area (Australia; Alderman and Skinner, 1957; Von der Borch et al., 1976,
Rosen et al., 1989; Warren et al., 1990). While both types of fluid become concentrated
during evaporation and, perhaps, modified by the precipitation of carbonates and evaporites,
giving rise to abundant dolomite formation, it remains unclear which mechanism prevailed on
the Carnian platform.

The Travenanzes Fm. differs from these potential modern analogues in its large amounts of

clay. In fact, dolomites occur in the Travenanzes Fm. as beds intercalated in a 100-m-thick
sequence of red clay, deposited on a distal alluvial plain, presumably under seasonally wet
conditions. This facies shows, except for the horizons containing marine fossils, striking
similarity to a Germanic Keuper facies, which represents an extended and entirely continental
playa lake system, also showing intercalations of primary dolomite in red clay (Reinhardt and



Ricken, 2000). Although the Travenanzes Fm. is clearly located, palaeogeographically, in the
Tethyan depositional region (Breda and Preto, 2011), its facies separation from a Germanic
Keuper facies may not be precisely coincident with palaeogeographic features, such as the
Vindelician high zone. We suggest that the composition and origin of ionic solutions
conducive to primary dolomite formation, either from continental water or seawater, is also an
indicator for the palaeogeographic separation between the two facies zones.
Here we provide a detailed investigation of dolomites of the Travenanzes Fm. to
reconstruct the processes and factors conducive to dolomite formation. We specifically
searched for sedimentary structures indicating that the initially deposited authigenic carbonate
was still unlithified, as it would be expected if it spontaneously precipitated from the shallow
water bodies of ephemeral lakes or tidal ponds. Radiogenic Sr isotope ratios ($^{87}Sr/^{86}Sr$) were
measured in the dolomites and compared with the known Triassic seawater Sr-isotope curve
(Veizer et al., 1999; McArthur et al., 2012) to determine if the ionic solutions conducive to
dolomite formation are derived from seawater or from continental runoff. Values were also
compared to dolomites from modern environments and to dolomites of clear continental
origin from the Germanic Keuper. Based on the new insights we discuss possible scenarios of
dolomite formation that could have prevailed along the Triassic western Tethys margins and
in similar evaporative environments.

**2 Geological setting**
The Dolomite mountains (Southern Tyrol and Venetian Alps; Fig. 1a) are well known for
their characteristic peaks consisting of Triassic carbonate platform limestones and dolomites.
These platforms developed all along the margins of the western Tethys ocean (Stampfli and
Borel, 2002), and are separated by deep basins in the middle Triassic and form an extended
coastal plain during the Carnian and Norian. The Adriatic plate rotated by almost 90° counter
clockwise during alpine orogeny (Ratschbacher et al., 1991; Handy et al., 2010). As a result,



the deep-water environments are found to the north in today's tectonic position although they
were originally located to the east (Fig. 1a). In the Dolomites, the Triassic paleogeography
was largely preserved in spite of Alpine deformation because the Dolomites form a ca. 60 km
wide pop-up structure bound by the periadriatic line to the north and northwest and the
Valsugana fault to the southeast (Fig. 1a, inset). Therefore, the Dolomites were never buried
to greater depth and have not experienced a metamorphic overprint (Doglioni, 1987). The
colour alteration index of conodonts in the Heiligkreuz Fm., underlying the Travenanzes Fm.
in this region, is 1, suggesting maximum burial temperatures of less than 50°C which are
confirmed by biomarker data (Dal Corso et al., 2012).

The Travenanzes Fm. lies unconformably above the Heiligkreuz Fm., and is overlain by

the Dolomia Principale (Hauptdolomit) with a transgressive boundary (Fig. 1b). Presumably
as a result of a change in climate and increasing humid episodes during the Carnian, large
amounts of siliciclastic material were deposited, entirely filling the more than 100 m deep
basins between the carbonate platforms of the Cassian dolomite (Gattolin et al., 2013; 2015).
These basin-filling deposits form a coastal succession or mixed carbonate siliciclastic ramp,
including large clinoforms with sandstones and conglomerates (Heiligkreuz Fm.; see Preto
and Hinnov, 2003; Gattolin et al., 2013; 2015). The overlying Travenanzes Fm. was deposited
on an extremely flat topography, as it consists of ca. 100-m-thick red and green claystone with
intercalated dolomites, evaporites and siliciclastic beds (Fig. 2; Kraus, 1969; Breda and Preto,
2011). In a south-north transect, it shows a typical interfingering between alluvial deposits
with conglomerates and sandstones to the south and a carbonate-dominated peritidal to sabkha
facies to the north (Breda and Preto, 2011). The upper boundary to the Dolomia Principale is
time-transgressive, i.e., it becomes younger from north to south. The Travenanzes Fm.
consists of three transgressive-regressive cycles, with the highstand deposits showing
identical peritidal carbonate facies as the Dolomia Principale (Breda and Preto, 2011). The



boundary to the Dolomia Principale is defined by the last occurrence of siliciclastic material
(Gianolla et al., 1998).
The depositional environment of the siliciclastic facies in the Travenanzes Fm. has been
interpreted as a dryland-river system by Breda and Preto (2011). Such a system occurs in arid
environments if rivers drain into a coastal alluvial plain, but do not reach the coast.
Evaporation along the way may lead to the formation of playa lakes, whereas on the seaward
side extended evaporative tidal areas, i.e., sabkhas, develop. Both types of environment are
well known for giving rise to modern dolomite formation (see references above). As the
Southern Alps were located in tropical latitudes, a warm arid climate, perhaps influenced by a
monsoon effect, had developed (Muttoni et al., 2003). Rivers provided large amounts of clay,
becoming partially oxidized under subaerial conditions, a typical red and green clay
succession containing palaeosols developed.
This facies is widespread throughout the Alpine and Tethyan realm during the Carnian, but
the same deposits are strongly deformed by alpine tectonics in most Austroalpine units,
forming a characteristic band of rauhwacke, the "Raibl beds" (e.g., Czurda and Nicklas,
1970). In the Travenanzes Fm. the entire sequence still shows its depositional architecture,
providing a pristine archive to study the diverse intercalated dolomites.
The Carnian and Norian deposits of the Keuper in the endorheic Germanic Basin show a
similar facies as the Travenanzes Formation. The Germanic deposits are described in more
detail by Reinhardt and Ricken (2000; and references therein), and they clearly represent
continental playa lake deposits. Here they are only included for comparison with the
Travenanzes Formation.

**3 Methods**
**3.1 Petrographic and mineralogical analysis**





A total of 39 hand specimens were collected from the stratigraphic section at Rifugio
Dibona 5 km west of Cortina d'Ampezzo (46.532727N/12.067161E; Fig. 1; Breda and Preto,
2011). Additional samples of Triassic dolomites from the Germanic Basin (Weser Fm. and
Arnstadt Fm. near Göttingen, Northern Germany) and modern dolomite from the Coorong
Lagoon (South Australia) and Deep Springs Lake (California) were also analysed for
comparison. Polished thin sections were carbon coated for analysis under the scanning
electron microscope (SEM) using a FEI Inspect S-50 SEM (Thermo Fisher Scientific,
Bremen, Germany). Element contents were determined semi-quantitatively using an EDX
detector (EDAX Ametek, New Jersey, United States) under high vacuum, a spot size 5.0 and
12.5 kV beam voltage at a working distance of 10 mm. Differences in mineralogy at the
micron scale were mapped in backscatter mode with high contrast.
For bulk mineralogical analysis three dolomite samples were milled with a disk mill. Clay
mineralogy was determined on 40 g aliquots that were leached two times for 24 h in 250 ml of
25% acetic acid to dissolve all carbonate (Hill and Evans, 1965). The clay mineral separates
were washed three times with $H_2O$ and centrifuged. The grain size fraction <2 μm was
collected by sedimentation in an Atterberg cylinder after 24 h 33 min. Oriented samples were
prepared by pipetting the suspensions (10 mg clay/ml) on glass slides and analysed after air
drying. To identify expandable clay minerals, the samples were additionally saturated with
ethylene-glycol and heated to 550°C (Moore and Reynolds, 1997). X-ray diffraction analysis
of bulk samples and clay mineral separates was performed with a PANalytical X'Pert Pro
diffractometer using CuKα radiation with 40 kV and 40 mA. The samples were scanned from
1.76° to 70° 2ϑ with a step size of 0.0167° and 5 s per step. The X-ray diffraction patterns
were interpreted using the Panalytical software "X'Pert High score plus" and Moore and
Reynolds (1997) for the clay minerals.
Total organic carbon (TOC) and total inorganic carbon (TIC) contents were determined for
seven samples of pure claystone, not containing any dolomite layers or nodules. This material




was used as carbonate-free control for acid leaching experiments as explained below. Ca. 0.2
g of dry sample powder was measured in a LECO RC-612 multiphase carbon analyser, at the
Department of Environmental Geosciences at the University of Vienna, with a temperature
ramp of 70°C per min to a maximum temperature of 1000°C.

**3.2 Carbon and oxygen isotope analysis**
Carbon and oxygen isotopes were measured on 28 samples which where micro-drilled
from thin section cuttings (see below). The samples were analysed with a Delta V Plus mass
spectrometer coupled to a GasBench II (Thermo Fisher Scientific, Bremen, Germany) at the
ETH Zürich (Zürich, Switzerland) following the procedure described in Breitenbach and
Bernasconi (2011). The precision was better than 0.1‰ for both isotopes. The oxygen isotope
values were corrected for kinetic fractionation during dissolution of dolomite in anhydrous
phosphoric acid at 70°C, using a fractionation factor of 1.009926 (Rosenbaum and Sheppard,

1986).


**3.3 Element analysis**
Total element concentrations were measured in leachates of the same three dolomite
specimen analysed by XRD and two claystones with the lowest inorganic carbon content. The
purpose of these measurements was to test the efficiency of the sequential extraction
procedure used for Sr-isotope analysis, and to determine potential origins of the Sr. The
samples were homogenized in an agate pestle and mortar and 100 mg of the homogenized
powder were weighed into centrifuge tubes. The samples were reacted in 10 ml 0.1 N acetic
acid and placed on a shaker for two days. The sample was centrifuged and the supernatant
was stored separately. The leaching step was repeated with 10 ml of 1 N acetic acid. Five ml
of each fraction were used for element concentration analysis (the rest was further processed
for Sr-isotope analysis; see below). The solutions were evaporated on a heating plate and the




residues were redissolved in 5 ml 2.5 N $HNO_3$. This step was repeated with 5 ml 5% $HNO_3$.
Concentrations were measured with a Perkin Elmer 5300 DV ICP-OES at the Department for
Environmental Geosciences (University of Vienna). Detection limits for the different
elements in rock (μmol/g) were: Al: 0.185, Ca: 0.025, Fe: 0.090, K: 0.026, Mg: 0.041, Mn:
0.002, Na: 0.004, P: 0.032, Ti: 0.002, Ba: 0.001, Sr: 0.001 and Rb: 0.012. The precision of the
measurements (relative standard deviation; RSD) for the elements Al, Ca, K, Mg, Ti, Ba and
Sr was ≤0.9% and for the elements Fe, Mn, Na, Rb, P it was ≤6.8%.

**3.4 Radiogenic Sr-isotope analysis**
To ensure that Sr from the pure dolomite phase is extracted specific areas free of clay
minerals were recognized by SEM and identified using an Olympus SZ61 microscope
equipped with a MicroMill sampling system (Electro Scientific Industries). Eleven samples
were drilled over a square area of 5-10 $mm^2$, or along a line in laminated rocks, to a depth of
350 μm. To prevent the powder from being blown away, the samples were drilled within a
drop of MilliQ-$H_2O$, and the suspension was transferred to a centrifuge tube using a pipette.
Also bulk samples, clay samples, pure celestine and barite purchased from W. Niemetz
(Servitengasse 12, 1090 Vienna, Austria), pure dolomite powder from Alfa Aesar (Thermo
Fisher – Kandel – GmbH, Postfach 11 07 65, 76057 Karlsruhe, Germany) and a fragment of a
single dolomite crystal were analysed as controls. They were crushed to a powder in an agate
mortar and pestle. Dolomite, barite, and celestine were mixed in a similar ratio as they occur
in the dolomites of the Travenanzes Fm. and run through the entire procedure as a control of
extraction efficiency. 14 mg of rock powder was weighed out for isotope analysis.
As additional precaution to extract the most pure dolomite phase for Sr-isotope analysis, a
sequential extraction was used. The extractions were routinely performed in 2 ml or 15 ml
polypropylene tubes with cap at room temperature on a shaker for 10 min to 24 h. The
following leaching reagents (always 2 ml) were used: 1 M NaCl, 3.3 M KCl, 0.1 N acetic



acid, 1 N acetic acid and 6 N HCl. Each reaction step was repeated once, and the residues
were washed with 2 ml of MilliQ $H_2O$ after each step to remove remains of the previous
solvent.
Sr was separated from interfering ions (e.g. Fe, K, Rb and Ca) using an ion exchange
column packed with BIO RAD AG 50W-X8 resin (200-400 mesh, hydrogen form). Leachates
were evaporated, dissolved in 6 N HCl and 2.5 N HCl and loaded to the column in 2 ml 2.5 N
HCl. Then 51 ml of 2.5 N HCl were run through the column to wash out the interfering ions.
The Sr was eluted with a further 7 ml 2.5 N HCl and dried after collection. Total procedural
blanks for Sr were <1 ng and were taken as negligible (the amounts of strontium in the
samples were always higher than 100 ng).
The isotopic composition of Sr was measured with a Triton (Thermo Finnigan) thermal
ionisation mass spectrometer. Sr fractions were loaded (dissolved in 1 µl $H_2O$) as chlorides
and vaporized from a Re double filament. The double filament configuration was used to
accelerate detachment of the Sr from the filament. The cup configuration was calibrated such
that masses 84, 85 (centre cup), 86, 87 and 88 are detected. The NBS987 Sr isotope standard
(number of replicates = 40) shows a $^{87}Sr/^{86}Sr$-ratio of 0.710272 ±0.000004 during the time of
investigation, with the uncertainty of the Sr isotope ratios quoted as 2σ. Interference with $^{87}Rb$
was corrected using a $^{87}Rb/^{85}Rb$ ratio of 0.386. Within-run mass fractionation was corrected
for $^{86}Sr/^{88}Sr = 0.1194$.

**4 Results**
**4.1 Petrographic description of dolomites**
Fig. 2 shows the distribution of the different types of dolomite through the 100-m-thick
lower, clay-rich interval of the Travenanzes Fm., above which the facies switches sharply to
massive bedded dolomites similar to those of the overlying Dolomia Principale.
Macroscopically three types of dolomite can be distinguished: homogenously bedded



dolomite, laminated dolomite, and nodular dolomite (Fig. 3a-c). The lower and middle part of
the clay-rich series harbours mainly homogeneous dolomite beds in red clay. Between 40 and
70 m several horizons with gypsum nodules occur (Fig. 3d). A 30-cm-thick fluvial
conglomerate with dolomite-cemented quartzarenites and pebbles of ripped up micritic
carbonate occurs at 75 m (Fig. 3e), above which palaeosols with dm-scale vertical peds,
possible root traces showing green reduction haloes, and nodular dolomite (calcic vertisols;
cf. Cleveland et al., 2008), are more frequent (e.g., Fig. 3b). Tempestite beds with
megalodonts, foraminifers and ostracods occur at 65 and 89 m. A pronounced transition
occurs in the uppermost ca. 8 metres of the clay-rich interval (Fig. 2b), where the clay entirely
changes from red to grey colour (Fig. 2c), and laminated dolomites become predominant
while evaporites and palaeosols are absent. The laminated dolomites (Fig. 3c) and cm- to dm-
scale dolomite-clay interlayers show intense slumping and soft sediment deformation and
pseudo-teepee structures (Figs. 3f, g). Here we provide a short summary of the petrographic
analysis of thin sections of the different types of dolomite with the most important features
compiled in table 1.

*Homogenous dolomites*

Homogeneous dolomite beds are usually 10 cm to 50 cm thick, embedded within clays and

with sharp, plane-parallel joints. They consist of dolomicrite, which was described as
aphanotopic dolomite by Breda and Preto (2011), according to the extended nomenclature for
dolomite fabrics by Randazzo and Zachos (1983). The sediment is matrix-supported and
contains irregular, partially rounded mud clasts (intraclasts) that consist of an aphanotopic
dolomite as the matrix. Some of the mud clasts contain smaller and somewhat darker
mudclasts or peloids (Fig. 4a, arrow). Soft sediment deformation is often not clearly visible
due to the homogeneous structure of the mud, but it can be observed where the mud clasts are
deformed within the matrix (Fig. 4b). Some of the homogeneous beds in the lower part of the



section show sub-millimetre lamination that is only visible under the microscope, where it
appears as an alternation of light (locally coarser) and dark aphanotopic dolomite.
The clay content in the homogeneous beds is generally low. A few beds (e.g. at 33.5 m in
the section) consist of silty or sandy dolomite, as reflected in a high abundance of detrital
quartz in thin section. Pseudomorphs after gypsum occur in a dolomite bed at 120 m (Fig. 4c,
d). Moldic porosity occurs in three layers at 43, 65 and 89 m, within aphanotopic dolomite.
These are the tempestite beds observed in the outcrop (cf. Breda and Preto, 2011).
One homogenous dolomite bed located at 64 m in the section contains oolithic grainstone,
lacking both an aphanotopic and a cement matrix (Fig. 4e). Ooids are either hollow (where
the cores may have been dissolved) or filled with sparite and are surrounded with an
isopachous cement rim.

*Nodular dolomites*

Nodular dolomites (Fig. 3b) often occur in beds of vertical peds linked to palaeosols as
indicated by horizons of typical vertical cracks showing green alteration fronts. Single
nodules also may sporadically occur embedded within metre-thick beds of red and green clay.
Nodules are usually 5 to 10 cm in size, consist of aphanitic dolomite or occasionally
somewhate coarser microspar, and in cross section show both red and pale areas. Most
nodules also show a deformed or brecciated internal structure with the interstices between the
clasts mostly consisting of matrix and clay cutans.

*Laminated dolomites*

Laminated dolomites occur in the upper part of the clay rich interval, between 90 and 110
m in the section (Fig. 4f-i). In the field, they show an alternation of light grey dolomite
laminae and dark grey to black clay laminae in the mm-range. Some dolomite laminae show
upward bending reminiscent of pseudo-teepee structures (Fig. 4f), and the space within the



teepee is sometimes infilled with sparry cement. Also, the bending of the laminae towards the
upward directed cuspids is reminiscent of load structures (dish structures), but they also may
represent desiccation cracks. The laminae are frequently ripped apart and fragments of
laminae occur reworked as flat pebbles embedded in an aphanotopic dolomite matrix (Fig.
4g). Some laminae show a microsparitic appearance and laminar fenestral porosity. In some
laminae a peloidal fabric is observed (e.g in Fig. 4f). Laminae are typically graded, whereby
the upper part is darker, indicating an increase in the clay content (Fig. 4h, i). The top of the
laminae is often truncated by an erosion surface, and rip-up clasts of the fine mud (mud
clasts) are embedded in the overlying coarse layer. Some laminated dolomites contain
continuous layers with inclusions of celestine crystals in the 100-μm-range, some of them
with barite in their centre (Fig. 5a-c). Occasionally also pyrite occurs.

Under the SEM, laminated dolomites show an anhedral structure in the 1-5 μm range. No

difference in mineral structure and grain size is usually observed between mud clasts and the
surrounding, often lighter-coloured matrix. Dolomite crystals at the margins between
dolomite and clay interlayers often coalesce into 5-μm-scale round aggregates consisting of
several subhedral crystals with different orientation (Fig. 6a, b; the crystals show orientation
contrast under BSE mode). Dolomite crystals are often porous, showing a somewhat
disordered appearance, but they are surrounded by syntaxial rims. In most cases, the rims
entirely fill the intercrystalline space, forming almost hexagonal compromise boundaries (Fig.
6c, d). These rims occur both in homogeneous and laminated dolomites.

*Germanic Keuper dolomites*

A sample from the Carnian Lehrberg Beds (middle Lehrberg bed; clay pit Friedland, 12

km south of Göttingen, Northern Germany; Seegis, 1997; Arp et al., 2004) shows a brittle
structure with high porosity. The material consists mainly of packed ooids or rarely peloids in



a sparitic cement matrix. Under the SEM, subhedral to euhedral dolomite in the 5-μm-range
are observed within the ooids (not shown).

A sample from the Norian Arnstadt Fm. (formerly termed "Steinmergelkeuper"; middle

grey series; locality of Krähenberg, 11 km SSW of Göttingen, Northern Germany; Arp et al.
2005) shows a mm-scale lamination and cm- to dm-sized laminated clasts, which were
interpreted as a stromatolite breccia. The laminae contain abundant agglutinated siliciclastic
grains (mainly quartz, subordinate albite) and phosphoritic fish scales. The dolomicrite shows
a subhedral structure in the ≤5 μm range with a few larger subhedral grains resulting in a
porphyrotopic fabric (not shown).

**4.2 Mineralogy**

Bulk dolomite shows a position of the 104 peak at a mean d-value of 2.88816 Å (Fig. 7a;

Table 2). This indicates a Ca content of 50.7%, based on the equation of Lumsden (1979).
The structural order is indicated by the ratio of the superlattice-ordering peak at (015) to the
(110) ordering peak. The height ratio is 0.44, which is near to 0.519 (Table 2) indicated for an
ordered dolomite in the Highscore database.

Clay mineral analysis (Fig. 7b-d) revealed illite in samples TZ14-1 and TZ14-7 and an R3

ordered illite-smectite mixed-layer clay mineral in sample TZ14-9. In the ethylene-glycol-
saturated state, the broad shoulder at 11.4 Å contains components of the illite 001 reflection
and of the fourth order of a 47 Å superstructure peak whose unit cell consists of three 10 Å
illite layers and one 17 Å smectite layer (Moore and Reynolds, 1997). This smectite
component was not observed in samples TZ14-1 and TZ14-7.

**4.3 Carbon content**

Total carbon contents in shales (Table 3) range from 0.06 to 0.51 wt%. Samples TZ16-1

und TZ16-19B showing the lowest TIC of 0.02 wt% were selected as controls to test for



$^{87}$Sr/$^{86}$Sr-ratios of Sr potentially adsorbed to clay minerals. TOC-contents are in the range of
0.05 - 0.16 wt%. Max. TIC-values are 0.46 wt%.

**4.4 Carbon and oxygen isotopes**
Carbon isotope values vary between -3.38 and +4‰ VPDB. Oxygen isotope values are
between -0.7 and +0.9‰ VPDB (three outliers show values as low as -1.5‰ VPDB; Table 4;
Fig. 8a). A clear distinction occurs between nodular dolomites showing negative $\delta^{13}$C-values
and homogeneous dolomites showing positive values. Laminated dolomites show
intermediate values and low variability. The oxygen isotopes show an upward increasing
trend (Fig. 8b). The calculated temperature of formation assuming a Triassic seawater
composition of -1‰ VSMOW using the fractionation equation of Vasconcelos et al. (2005)
shows temperatures between 29 and 39°C. A more positive value of the water would result in
higher temperatures.

**4.5 Element composition of the dolomites**
Concentrations of the elements Al, Ca, Fe, K, Mg, Mn, Na, P, Ti, Ba, Sr, and Rb (mmol/g
sample) are shown in Table 5. Ca contents are between 1.68 and 2.33 mmol/g in the 0.1 N
acetic acid fraction and between 2.71 and 2.87 mmol/g in the 1 N acetic acid fraction. Mg
contents are between 1.61 and 2.34 mmol/g in the 0.1 N acetic acid fraction and between 2.48
and 2.64 mmol/g in the 1 N acetic acid fraction. Based on these concentrations, the amount of
dolomite dissolved is between 30 and 43 wt% of the bulk sample in the 0.1 N acetic acid
fraction and between 49 and 52 wt% in the 1 N acetic acid fraction of the sequential
extraction. In total, between 84 and 90 wt% of the bulk sample were dissolved during these
two extraction steps. If molar concentrations of Ca are plotted vs. Mg a linear trend with a
slope of 0.935 is observed (Fig. 9a), indicating 48.3 mol% $MgCO_3$ in the dolomite phase.

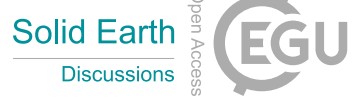

The Sr-concentrations in bulk dolomite samples are in the range of 0.38 and 1.16 μmol/g
in the 0.1 N acetic acid fraction and between 0.57 and 0.79 μmol/g in the 1 N acetic acid
fraction (except one extremely high value of 34.91 μmol/g in sample TZ14-9). These contents
are much higher than in pure clay mineral samples with 0.047-0.417 μmol/g in the 0.1 N
acetic acid fraction and even lower concentrations (<0.19 μmol/g) in the other fractions. In
all samples measured by ICP-OES, rubidium (Rb) concentrations are below the detection
limit of 0.012 μmol/g.
Correlation of Sr contents to other elements did not show clear trends. In particular, Sr-
content did not correlate with Mg or Ca. Sr correlates with K (Fig. 9b), but at the same time,
K is extremely low in all clay mineral leachates.

**4.6 Sr-isotopes**
$^{87}Sr/^{86}Sr$-ratios of pure minerals
Results of Sr-isotope measurements are listed in Table 6. Repeated extractions of
chemically pure dolomite reference material dissolved in 0.1 N acetic acid showed a range of
$^{87}Sr/^{86}Sr$-ratios between 0.709942 ±0.000011 and 0.710831 ±0.000007. Pure single crystals of
dolomite extracted sequentially showed the highest value (0.708401 ±0.000040) in the 1 M
NaCl fraction. Values in the 0.1 N acetic acid fraction (0.707735 ±0.000006) and the 1 N
acetic acid fraction (0.707666 ±0.000006) are lower by almost 0.001 than in the NaCl
fraction.
In pure barite, $^{87}Sr/^{86}Sr$-ratios decrease by about 0.0013 in the extraction sequence from 0.1
N acetic acid to 6 N HCl. Celestine is highly soluble and was only measured in the 1 M NaCl
fraction and one time in 0.1 N acetic acid. It shows similar values as in the 1 M NaCl fraction
of the pure barite-celestine-dolomite mixture (0.708038 ±0.000003), but the latter increased to
0.709501 ±0.000040 in the 0.1 N acetic acid fraction.




*$^{87}$Sr/$^{86}$Sr-evolution during sequential extraction of dolomites of the Travenanzes Fm.*

Different modifications of the sequential extraction were investigated using three samples (TZ14-1, TZ14-7 and TZ14-9; Table 6). $^{87}$Sr/$^{86}$Sr-ratios decrease in sample TZ14-1 from 0.708125 ±0.000012 to 0.707666 ±0.000004 with increasing strength of the leaching reagent, while the values remain almost constant in sample TZ14-9. However, repeating the 0.1 N acetic acid extraction (for 36 h) after a rather intense first extraction (4h, 12h, 4h) resulted in extremely high values (0.715417 ±0.000250 in TZ14-1 and 0.7192266 ±0.000455 in TZ14-9). Standard deviations are also higher than in the other fractions.

The sequential extractions were repeated, whereby the Sr-concentrations were determined by ICP-OES (see section above). In addition, 1 N acetic acid and 6 N HCl fractions were extracted. Results are similar to the previous extraction sequences, but the values further decreased in the 1 N acetic acid fraction. Only the HCl-fraction showed very high values of 0.730453 ±0.000005 in sample TZ14-7.

Sequential extractions of the clay samples from the Travenanzes Fm. show a similar increase with the sequential extraction steps from 0.1 N acetic acid fraction to 6 N HCl, where $^{87}$Sr/$^{86}$Sr-ratios reach similar values as in the dolomite extracts (from 0.722998 ±0.000018 to 0.733910 ±0.000024).

*$^{87}$Sr/$^{86}$Sr-ratios in micro-drilled dolomite*

Eleven dolomite samples were micro-drilled from areas where dolomite was most pure based on examination by SEM and dissolved in 0.1 N acetic acid. The values of the Travenanzes Fm. are in the range of 0.707672 ±0.000003 to 0.707976 ±0.000004. The highest value occurs in a dolomite nodule, while no systematic difference between homogenous and laminated dolomite was observed. Dolomite of the Germanic Keuper samples shows much higher $^{87}$Sr/$^{86}$Sr-ratios of 0.709303 ±0.000006 and 0.709805 ±0.000005, respectively.



*$^{87}$Sr/$^{86}$Sr-ratios of modern dolomites (Deep Springs Lake, Coorong Lakes)*

Dolomites of Deep Springs Lake show strongly radiogenic values of 0.713086 ±0.000004 and 0.713207 ±0.000004, which are much higher than modern seawater values, showing a $^{87}$Sr/$^{86}$Sr-ratio of 0.709234 ±0.000009 (DePaolo and Ingram, 1985). In contrast, dolomite from the Coorong Lakes (Milne Lake) shows ratios between 0.709251 ±0.000004 and 0.709275 ±0.000003, which is very near to modern seawater. Different incubation times (5 min und 10 h) in 0.1 N acetic acid had no influence on the isotope ratios.

## 5 Discussion

### 5.1 Interpretation of microfacies within different types of dolomite

*Homogeneous dolomite beds*

The homogeneous dolomite beds, which are mainly intercalated in the lower, clay-rich part of the Travenanzes Fm., consist of fine-grained dolomicrite (aphanotopic dolomite), with occasional intraclasts of the same aphanotopic dolomite as the matrix. Soft sediment deformation and dolomicrite infill between mud clasts indicate that this sediment consisted to a large extent of unlithified carbonate mud. Based on the abundance of fine mud, water energy was probably not very high (Demicco and Hardie, 1994), although reworking and partial rounding of the mud clasts require at least occasionally higher water energies. According to the standard microfacies concept this type falls into SMF 23 ("non-laminated homogeneous micrite and microsparite without fossils"), indicating a deposition in "saline and evaporative environments, e.g. in tidal ponds" (Flügel, 2010). Also, SMF 24 ("lithoclastic floatstones, rudstones and breccias") is observed in some of the beds where mud clasts are abundant. These facies types are consistent with supersaturation-driven precipitation of a fine-grained authigenic carbonate in environments partially restricted from open seawater and would match both with a coastal sabkha environment and/or with shallow ephemeral lakes.



Ephemeral lakes may have formed on extended coastal alluvial plains along the Tethyan
margin during the Carnian. The fine mud may have been homogenized and redistributed due
to minor wave action in the ponds (cf. Ginsburg, 1971).

In a semi-arid climate, episodic flooding of the alluvial plains by river water, which

however mostly evaporated before reaching the coast, led to the formation of a dryland river
system (Breda and Preto, 2011). The fluvial system may have supplied water to temporally
existing evaporating ponds. Alternatively, the alluvial plain may have been sporadically
flooded by seawater, explaining the intercalations of authigenic dolomite layers in the
succession of alluvial clays. Homogeneous dolomites show a positive carbon isotope
signature between 0.7 and 4‰ VPDB (except one outlier), which would be consistent with
formation from unaltered marine carbon in evaporative brine, with no significant contribution
of $^{12}C$ derived from organic matter. As indication of evaporative conditions, several gypsum
beds occur between 45 and 70 m in the section, and pseudomorphs after gypsum were
observed in a thin section of a dolomite at 120 m (Fig. 4c, d). But evaporites may not always
be preserved as they were most likely dissolved due to seasonally wet conditions.

While most homogeneous dolomite beds consist of aphanitic dolomite, a bed of dolomitic

ooid grainstone devoid of matrix occurs at 64 m (Fig. 4e), and tempestites showing moldic
porosity indicative of dissolved allochems and dissolved fossils occur at several levels in the
section. These beds must represent events of higher water energy, contributing sediment from
more open marine areas. The presence of marine fossils, such as *Megalodon*, indicate that at
least episodically the environment was marine influenced. The microfacies of the oolite falls
into SMF 15, which indicates proximity to the seaward edge of the platform. A similar facies,
however, is encountered in the Carnian Lehrberg Beds (Seegis, 1997) in a lacustrine setting.
Several beds containing abundant siliciclastic material (mainly angular quartz clasts) are more
likely due to a riverine flooding event, providing detrital material from the continent. Thus,



the microfacies in homogenous dolomite beds indicates both marine and continental influence
on the depositional environment.

*Laminated dolomite*
In the upper part of the clay-rich interval, predominantly laminated dolomites reminiscent
of loferites (Fischer, 1964) occur. The change from more homogeneous to laminated dolomite
intercalations correlates with the change from red to dark grey clay. The lamination consists
of millimetre-scale dolomite/clay interlayers suggesting an alternation of clay and fine
dolomite deposition. The microfacies falls into SMF 25 ("laminated evaporite-carbonate
mudstone facies") indicating an "upper intertidal to supratidal sabkha facies in arid and
semiarid coastal plains and evaporitic lacustrine basins" (Flügel, 2010). Laminae showing soft
sediment deformation could not be attributed to stromatolitic bindstone facies (SMF 19 to 21).
Only some layers showing a coarser fabric with interstitial dolosparite or dolomicrosparite
containing putative peloids have been interpreted as microbial laminites (Preto et al., 2015).
Mostly, graded bedding indicates a direct sedimentation process rather than in situ
precipitation of the primary carbonate within a microbial mat (Vasconcelos et al., 2006;
Bouton et al., 2016; Court et al., 2017; Perri et al., 2018). A detrital origin of the clay in the
dolomites is confirmed by the well-ordered illite-smectite mixed-layer composition which
would be atypical for authigenic clay minerals. Frequent subaerial exposure and desiccation
may explain why the sediment was not homogenized and the lamination is preserved. This is
supported by the occurrence of pseudo-teepee structures as remnants of desiccation cracks.
Rip-up clasts were formed during subsequent flooding, whereby angular flat pebbles occur
where the sediment was desiccated or partially lithified. However, laminae also show
frequently plastic deformation (e.g. in Fig. 3g) where the mud was still unlithified.
Some uncertainty exists as to whether the facies was peritidal or represents ephemeral
lakes, as suggested for the homogeneous dolomites above. Episodic high water-energy





indicated by the rip-up clasts, combined with frequent desiccation, could point to evaporative
tidal conditions, as they occur in a sabkha. What is atypical for a modern sabkha is the large
amount of detrital input. But this is owed to the seasonally wet conditions during the Carnian
and the facies can be considered a mixed facies of alluvial plain and coastal sabkha: a "dirty"
sabkha. Under such conditions, the large amounts of evaporites, in particular gypsum, as they
usually occur in a sabkha, could have been dissolved. Why the occurrence of laminated
dolomites coincides with the transition from red to grey clays is not clear but may be related
to more permanently water-saturated conditions in the subsurface, while the surface was
exposed to periodic desiccation. Also this would be consistent with a sabkha environment.

*Nodular dolomite*
The clay beds were subject to strong evaporation and vadose diagenesis causing oxidation
and red colour. This generally indicates, at least seasonally, arid conditions. Dolomite nodules
that occur sporadically within certain intervals show internal brecciation, which probably
occurred after sedimentation. Internal brecciation is a typical feature of present day calcretes
in arid environment (e.g. Mather et al., 2018). Slightly negative $\delta^{13}$C-values indicate a
contribution of carbon derived from organic matter degradation, suggesting that they formed
within the sediment. Presumably the formation of dolomite nodules could be related to
diagenesis in palaeosols. In the upper part of the section (between 80 and 105 m) dolomite
nodules are associated with green reaction haloes along vertical peds in palaeosols of vertisol-
calcisol type (Preto et al., 2015). Carbonate formation may have been related to reducing
fluids in water-logged soils during humid intervals, while the crack formed during desiccation
in dry periods, perhaps facilitated by the presence of expandable clay minerals (smectite).

**5.2 The origin of ionic solutions conducive to dolomite formation**
Overall, the dolomites in the Travenanzes Fm. show facies that match a variety of potential





depositional environments. They show some similarity to the Germanic Keuper, and it is not
entirely clear from the facies, whether a marine influence occurred, except if indicated by
marine fossils, as in the tempestite beds. To better trace the origins of ionic solutions to the
environments that were conducive to dolomite formation, Sr-isotopes were analysed.

*Strontium derived from seawater*
Radiogenic $^{87}Sr/^{86}Sr$ ratios can be indicative of the source of ionic solutions from which the
dolomite precipitated (Müller et al., 1990a; Müller et al., 1990b). Sr-isotopes in selected
dolomites through the Travenanzes Fm. at the Dibona section showed values between
0.707672 ±0.000003 and 0.707976 ±0.000004. We correlate the Dibona section (Fig. 10) with
the Carnian seawater curve (Korte et al., 2003). Although the age interval of the Travenanzes
Fm. is not precisely constrained, findings of ammonites at the base of the succession suggest a
Tuvalian II age (*subbullatus* zone, 232.5-231.0 Ma; Ogg, 2012). The upper boundary of the
Travenanzes Fm. is time-transgressive and hence the age not precisely constrained. We
assume that the sedimentation rate was at least as high, or higher, than in the peritidal
carbonates of the Dolomia Principale. In this region, the Dolomia Principale includes a part of
the Rhaetian (Neri et al., 2007) and, thus, its upper boundary is near the Triassic-Jurassic
boundary at 201.3 Ma. The seawater curve was fixed at the lower boundary of the
Travenanzes Fm. and the time axis was varied to fit the seawater curve parallel to the
envelope of minimal $^{87}Sr/^{86}Sr$-ratios measured in the dolomites (Fig. 10). The base of the first
massive dolomite at 110 m in the profile would then have an age of approximately 229 Ma.
Comparison with the seawater curve shows that the dolomites of the Travenanzes Fm. have
largely marine $^{87}Sr/^{86}Sr$-ratios (Fig. 10). Only values from micro-drilled samples most gently
extracted with 0.1 N acetic acid were used for this reconstruction, and the resulting values all
lie within a range of 0.00022 with the seawater values (grey shaded area). This scatter
towards more positive values, compared to seawater, may be due to a small influence of





594 continental water. Indeed, during deposition of the Travenanzes Fm. sufficient continental

595 water would have been available from rivers, and ions may have become concentrated while

596 the water was evaporating in the distal alluvial plain. Alternatively, Sr desorbed from clay

597 minerals could have added more radiogenic values to the brine. But even if a small influence

598 of Sr of continental origin is present, because of the much higher Sr concentrations in

599 seawater, the marine signal is dominant.

600  This observation does not support the classical Coorong model for dolomite formation,

601 where alkalinity is largely derived from continental groundwater. The Coorong Lakes in

602 South Australia are ephemeral lakes largely supplied with groundwater (Von der Borch et al.,

603 1975). Strangely, though, the $^{87}Sr/^{86}Sr$ ratios we measured in Milne Lake (one of the Coorong

604 Lakes) show modern seawater composition (Fig. 11), but this can be explained as the local

605 groundwater largely originates from a Pleistocene carbonate aquifer, accordingly carrying a

606 Pleistocene Sr-isotope signature. A similar scenario for the Travenanzes Fm. is unlikely as the

607 only large-scale preceding carbonate platforms at that time were the Late Ladinian-Carnian

608 Cassian dolomite platforms (Russo et al., 1997). But based on the stratigraphic context, all

609 basins between these platforms were infilled by the Heiligkreuz Fm. and an extremely flat

610 topography had established that was stratigraphically overlain and sealed by the alluvial

611 deposits of the laterally persistent Travenanzes Formation. Furthermore, the Travenanzes Fm.

612 consists of 100 m of impermeable clay (containing expandable clays) such that a long-

613 distance transport of groundwater can be excluded.

614  We conclude that the $^{87}Sr/^{86}Sr$ ratios of the dolomites truly represent a dominating marine

615 influence. Presumably, seawater was transported to the interior of the platforms by episodic

616 flooding (spring tide or storm) events. Even in a seasonally wet climate the input of river

617 water on Sr-isotopes was insignificant compared to the influence of ions (including Sr) from

618 seawater that became concentrated by evaporation. Laminated dolomites in the uppermost

619 part of the section show values most similar to seawater composition, which is consistent with



a greater influence of peritidal conditions.

*The influence of Sr adsorbed to clay minerals*
An outlier with higher $^{87}Sr/^{86}Sr$ ratios occurs in a dolomite nodule, presumably representing
a more continental influence or perhaps more seasonally wet and evaporative conditions with
less marine influence. But also higher values may be due to contamination and partial
leaching of clay minerals within the dolomite samples. Within the extraction sequence (1 M
NaCl → 0.1 N acetic acid → 1 N acetic acid), the $^{87}Sr/^{86}Sr$ ratio generally remains constant or
becomes slightly less radiogenic, i.e., more similar to seawater. However, the values strongly
increase with leaching in 6 N HCl (Table 6). A modification of the $^{87}Sr/^{86}Sr$ ratios due to
contamination by $^{87}Sr$ from the radioactive decay of $^{87}Rb$ to $^{87}Sr$ can be considered as
negligible since the concentrations of Rb was below the detection limit of 0.05 ppm (Table 5)
and the half time of the decay is 48.8 billion years. Also, an influence of celestine and Sr-rich
barite, observed under the SEM, on the Sr-isotope values can be largely excluded. These
mineral phases are bound to distinct layers of the laminated dolomites, where they could be
avoided by micro-drilling in areas where the dolomite was pure. Only one value in sample
TZ14-9 shows extremely high Sr-concentrations. This sample was micro-drilled near to a
celestine layer and it is therefore not surprising that a celestine crytal may have been leached.
The isotopic composition of the celestine is also similar to Carnian seawater.
In the NaCl-fraction only minimal amounts of dolomite are dissolved. The slightly more
radiogenic $^{87}Sr/^{86}Sr$ ratio may be derived from Sr that is lightly adsorbed to clay minerals and
finely dispersed in the clay matrix, although $Sr^{2+}$ as a two-valent cation is more strongly
adsorbed to clay mineral than $Na^+$, and thus not easily desorbed by NaCl. With increasing
extraction efficiency and purity of the carbonate phase, the values approach seawater values
in the 1 N acetic acid fraction. Also, values from micro-drilled samples are generally more
similar to seawater values, probably because more pure dolomite was sampled (Table 6). 1 N



acetic acid is usually observed not to strongly attack interlayer ions in clay minerals.

Clay minerals leached in 6 N HCl show significantly more radiogenic values compared to

dolomite samples (Table 6). This finding is consistent with strongly radiogenic values in the 6
N HCl-fraction of dolomite samples (up to $0.730453 \pm 0.000005$) and supports that the clay
minerals are the carriers of a Sr-pool significantly more radiogenic than the carbonate phase
showing marine values. Sr is known to adsorb to illite-smectite mixed layer clay minerals
(Missana et al., 2008). The HCl-fraction most likely includes adsorbed Sr, and Sr occupying
the interlayer positions of the clay minerals, and presumably also structurally bound Sr in the
clay mineral phase. In particular, illite-smectite mixed-layer clay minerals, as detected by
XRD of the clay mineral separate in sample TZ14-9 (Fig. 7d), could have two different
origins, burial diagenesis and continental weathering. Based on the tectonic setting and low
burial depth of the Dolomites, burial depth for smectite-illite transition has not been reached.
Therefore, these minerals are most likely derived from silicate weathering, with the Sr-
signature representing the crustal origin of the parent rock. Our finding of radiogenic Sr-
isotope ratios supports that clay minerals did not essentially incorporate the Sr from seawater,
delivered at high sealevel stand. It is therefore clear that Sr extracted from the dolomites is not
derived from clay minerals.

*Dolomite as primary archive of Sr-isotope signatures*

The question is, whether Sr truly represents the conditions of dolomite formation or

whether it inherits the Sr content of some precursor phase. Baker and Burns (1985) and
Vahrenkamp and Swart (1990) showed very small distribution coefficients between aqueous
and solid solutions, and high Sr-contents measured in Abu Dahbi sabkha dolomites (Müller et
al., 1990b) may be derived from precursor aragonite. However, dolomite in the Travenanzes
Fm. is largely primary (Preto et al., 2015) and thus not formed from an aragonite precursor. It
is likely that remobilization of Sr during burial may have released parts of the Sr from



dolomite which is now present as celestine and barite inclusions.
Furthermore, Sánchez-Román et al. (2011) demonstrated that protodolomite forming in
culture experiments contain Sr in the range of several thousand ppm. The incorporation
mechanism of Sr is still not entirely clear, since Sr is a large ion that should occupy the sites
of Ca in the crystal lattice. However, in Sánchez-Román et al. (2011) Sr appears to correlate
with the Mg content, and another incorporation mechanism may occur, such as by surface
entrapment. A correlation of Sr-contents with K-contents is observed for the Travenanzes
dolomites. It could be circumstantial, but would not be inconsistent with an alternative
mechanism of Sr-incorporation, such as surface entrapment. Even if it is taken into account
that only protodolomite formed in microbial culture experiments (Gregg et al., 2015), natural
modern dolomites are often rich in Sr (e.g. Meister et al., 2007). The Sr could occur in
disordered nano-structural domains that are not picked up in the bulk XRD-signal.
Alternative, non-classical nucleation and growth pathways, e.g. by nano-particle attachment,
could play a role in the abnormal partitioning of Sr in the dolomite lattice. Thus, a high Sr-
content in the Travenanzes Fm. or in Abu Dhabi Sabkha dolomites is likely a true signature of
primary dolomites.

**5.3 Mode of dolomite formation and comparison with known models**
*Primary dolomite formation*
Several indications support that the origin of dolomite in the Travenanzes Fm. is largely
primary. Formation temperatures reconstructed from oxygen isotopes and assuming Triassic
seawater composition of -1‰ VSMOW are between 28 and 33°C. If a typical $^{18}O$ enrichment
of 3‰ in a sabkha (McKenzie et al., 1980; McKenzie, 1981) is assumed, the calculated
temperatures would be between 40 and 50°C, which is still within a range possible in a
sakbha. Both temperature and evaporation may have changed over time, which may explain
the observed linear trend in oxygen isotopes across the section (Fig. 8B). Furthermore, there is



no co-variation between $\delta^{13}C$ and $\delta^{18}O$, as opposed to evaporation in hydrologically closed
settings such as the Germanic Keuper basin (Reinhardt and Ricken, 2000; Arp et al., 2005).
Both oxygen isotopes and nano-crystalline structures observed by Preto et al. (2015) preclude
a later pervasive recrystallization during burial diagenesis. Sedimentary structures indicate
that most of the homogenous dolomite and laminae containing aphanotopic dolomite was
unlithified, and dolomite was therefore deposited as fine-grained mud. This is further
supported by mm-scale interlayering of clay and dolomite in the laminated dolomites near the
top of the sequence, and some dolomite/clay couplets showing a fining-upward bedding.
Based on the observation of nano-crystal structures, replacement did not take place and it
appears logical to assume that the primary phase was already dolomite.

While most of the dolomite may have been primary, micron-scale interstices between the

dolomicrite grains must have been cemented after deposition. This cementation resulted in
rims visible under the SEM and resulting in near hexagonal compromise boundaries. The
cement may have contributed $^{13}C$-depleted carbon during early diagenesis. The lowest $\delta^{13}C$
values of -3.4‰ VPDB occur in the nodules. There is no indication that these nodules formed
at the surface. They rather formed within the sediment, probably due to reducing conditions
and influenced by dissolved inorganic carbon from degrading organic matter in the
palaeosols. Homogeneous and laminated dolomites are clearly distinct from nodules in their
carbon isotope compositions (Fig. 8a), indicating only a minor contribution from pore-water
derived dissolved inorganic carbon. Carbon isotope values are thus largely consistent with a
primary precipitation. The mode of dolomite formation as fine mud and subsequent
cementation is comparable to several modern sites of dolomite formation.

*The sabkha model*

The classical sabkha model involves dolomite formation under intra-supratidal conditions,

concentration of brines through either seepage reflux (Adams and Rhodes, 1960) or





evaporative pumping (Hsü and Siegenthaler, 1969; Hsü and Schneider, 1973; McKenzie et
al., 1980; McKenzie, 1981) and precipitation of dolomite upon increase of the Mg/Ca ratio
due to gypsum precipitation (see Machel, 2004, for a more detailed discussion of varieties of
sabkha models). This group of models allow for a mixture of seawater and continental
groundwater, with seawater providing mainly the ions for dolomite precipitation. Coastal
sabkhas are typically characterized by laminated (Lofer-type) dolomites, whereby the laminae
are largely still unlithified after deposition (Illing, 1965; Bontognali et al., 2010; Court et al.,
2017). In fact, in the sabkha of Abu Dhabi, both pathways, via replacement of precursor
aragonite and by direct precipitation of dislocation-ridden primary dolomite, were observed
(Wenk et al., 1993).

The sabkha model is thus a reasonable model for the uppermost parts of the Travenanzes

section, showing laminated dolomites, marine Sr-isotope values and indications of frequent
desiccation and flooding in a peritidal setting. Yet, the conditions differed from the modern
sabkhas along the Persion Gulf by the large amount of alluvial clay (dirty sabkha), as opposed
to aeolian sand. Most of the fine lamination then may result from periodically varying
conditions, perhaps with clay deposition during episodes of fluvial discharge and carbonate
deposition during evaporative conditions.

*The continental playa lake model*

The playa lake model was originally suggested by Eugster and Surdam (1973) for dolomite

of the Green River Formation (Wyoming), but the primary formation of fine dolomite mud is
observed in many alkaline playa lakes, such as Deep Springs Lake (Peterson et al., 1963;
Clayton et al., 1968; Meister et al., 2011), Lake Acigöl (Turkey; Balci et al., 2017), Lake
Neusiedl (Austria; cf. Neuhuber et al., 2016), Lake Van (Turkey; McCormack et al., 2018; for
an overview see Eugster and Hardie, 1978, and Last, 1990). This type of setting has also been
suggested for the Germanic Keuper deposits during the late Carnian and Norian, when the



Germanic Basin was entirely disconnected from the Panthalassa ocean and was continental
(Reinhardt and Ricken, 2000). The Travenanzes Fm., with its homogeneous dolomite
intercalations in red and green clays, is strikingly similar to playa-lake Keuper facies in the
Germanic Basin. There, dolomite formed upon evaporation and concentration of the
continental brines under semi-arid climate.
However, Sr-isotope data support a dominantly marine origin of ionic solutions to the
Travenanzes Fm., whereas Sr-isotopes are strongly radiogenic in the Germanic Keuper
dolomites (or in Deep Springs Lake; Fig. 11). The two settings are thus fundamentally
different. Even dolomite nodules, showing somewhat more radiogenic values than seawater in
the Travenanzes Fm., are still indicating a dominating marine influence. The slightly more
radiogenic influence could be due to the clay minerals present in the nodules that were
difficult to entirely separate from the carbonate. Also, dolomite nodules may have formed in
relation to palaeosols, during somewhat more humid times and, thus, may have been slightly
influenced by continental water input from the rivers.

*The coastal ephemeral lake model (Coorong model)*
The Coorong model was proposed by Von der Borch et al., 1975; Von der Borch 1976;
Rosen et al., 1989; see Warren, 2000, for detailed information), explaining the formation of
primary and uncemented dolomite in the Coorong lakes of South Australia. The isotope
values show that the contribution of ionic solutions, and hence alkalinity, of continental origin
to the dolomitizing fluids was minimal and that the dolomites are seawater derived. This may
be distinct from the typical Coorong model, where alkalinity is provided from an inland karst
system. But other coastal ephemeral lakes exist, such as along the Brasilian coast north of Rio
de Janeiro. Partially unlithified dolomite occurs in Brejo do Espinho (Sánchez-Román et al.,
2009), which is in fact largely similar to the Coorong lakes, but ionic solutions are largely
derived from seawater.



A coastal ephemeral lake model would probably be most suitable to explain homogeneous

dolomite beds of the Travenanzes Fm., where hypersaline ponds may have formed in a

dryland river system. However, unlike the recent ephemeral lakes (such as Lagoa Vermelha,

Brejo do Espinho and the Coorong Lakes) the clay-rich sediment must have inhibited

groundwater flow. Hence, while modern coastal ephemeral lakes receive their water largely

through seawater percolating through porous dune sand, episodic flooding with seawater must

have provided ionic solutions for dolomite formation on the Carnian platform.

*A non-actualistic system*

Overall, the depositional environment reconstructed for the Travenanzes Fm. shows

similarities to modern systems were dolomite forms. Among all the scenarios, a coastal

ephemeral lake model would be most similar to the conditions conducive to homogeneous

dolomites, lacking signs of frequent desiccation, while a coastal sabkha model may explain

the laminated intervals near the top of the studied succession. In contrast to any modern

systems, the clay rich sediments of the Travenanzes Fm. precluded any transport of

groundwater, which plays a role for ionic transport in both the modern day ephemeral lake

model and the different versions of sabkha models. Although modern systems provide valid

analogues for the mechanism of dolomite formation in the past, and probably throughout

Earth's history, none of them is an exact environmental analogue. The Carnian alluvial plains

that covered an enormous area along the Tethys margin (e.g. Garzanti et al., 1995) represent a

non-actualistic system in terms of their sedimentary, hydrological and climatic conditions.

Besides, the geochemistry of Tethys seawater may also have been different from today, an

issue that requires further investigation. These aspects need to be taken into account if we

intend to understand the role of dolomite formation through Earth history.

In the light of a possible spontaneous precipitation as fine mud in the water column,

perhaps via formation and aggregation of nano-particles, further discussion of a nucleation

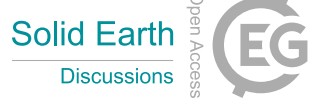

and growth pathway of dolomite will be necessary. While several modifiers may also play a
role in the water column, such as dissolved organic matter (Frisia et al., 2018), microbial EPS
(Bontognali et al., 2013), or suspended clay particles (Liu et al., 2018), fluctuating conditions
inducing spontaneous nucleation and growth of dolomite, in agreement with Ostwald's step
rule (Deelman, 1999), require further consideration as a factor favourable for dolomite
formation on a seasonally variable platform (Meister and Frisia, accepted).
The main finding of this study is that most of the dolomite in the >100 m thick
Travenanzes Fm. probably formed through direct precipitation in a seawater-derived solution.
This mode of primary dolomite formation has rarely been considered in the study of
geological dolomite bodies, but may explain the genesis of many other large-scale, fine
dolomite units that preserve fossils and sedimentary structures.

### 814 6 Conclusions

Dolomite beds intercalated in a 100-m-thick Carnian alluvial clay sequence in the
Travenanzes Fm. largely formed as fine-grained primary mud. The depositional environment
was minimally affected by currents and most likely prevailed as ephemeral lakes in an
extended alluvial plain or dryland river system. The large amounts of clay are related to at
least seasonally wet conditions. Also palaeosols and diagenetic dolomite nodules could have
formed under such conditions. The facies resembles strongly those of the Triassic playa lakes
prevailing in the Germanic Basin or in the modern Deep Springs Lake.
Sr-isotopes clearly show a marine signal, indicating seawater as the main source of ions.
The depositional environment shows most similarities with coastal ephemeral lakes resulting
in the deposition of homogeneous dolomite beds through most of the sequence, changing into
a "dirty" sabkha near the top of the sequence, where fine dolomite/clay interlayers suggest
alternating deposition of extremely fine authigenic dolomite from evaporating water, and
clay.



Overall, Sr-isotopes and petrographic observations provide insight into a non-
uniformitarian system including both elements of coastal ephemeral lake systems and sabkhas
as an environment of primary dolomite formation. Considering the precipitation of primary
dolomite from coastal lakes or ponds may help explaining other dolomite deposits with
preserved sedimentary features throughout geologic history.

*Acknowledgements*. We thank C. Beybel, I. Wünsche, and L. Slawek for preparing high-
quality petrographic thin sections. Thanks also to W. Obermaier for analysing element
concentrations by ICP-OES and P. Körner for support during TOC measurements. S.
Niebergall provided some of the petrographic images. We furthermore thank S. Viehmann for
help during sampling and supervision of the students in the field and B. Bethke for her strong
support in the laboratory. Thanks also to M. Lorencak for the help during sampling of
dolomite from the Coorong Lagoon. We thank S. Frisia for input and constructive criticism.
F. Franchi and H. Machel reviewed an early version of this manuscript. The study was
partially supported by the Marie Curie Intra-European Fellowship Project Triadol (Project no.

626025).

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

**Figure Captions**
**Figure 1. (a)** Palaeogeographic map of the Southern Alpine to Germanic domains during the
middle Triassic, reproduced from Brack et al. (1999; modified). Bal: Balaton; BG: Burgundy
Gate; Car: Carnian Alps; ECG: eastern Carpathian Gate; Lomb: Lombardy; NCA: Northern
Calcareous Alps; SMG: Silesian Moravian Gate. Inset: Tectonic map of the Southern Alps



(Brack et al., 1996, modified) showing the sampling location at Rifugio Dibona. GL:
Giudicarie Line; PL: Pustertal Line; VL: Val Sugana Line. (**b**) Stratigraphy of the middle to
late Triassic in Venetian Alps, showing a transition in geometries from basin and platform
topotraphy during the Lower Carnian to an extended alluvial to tidal plain in this Upper
Carnian. The shaded area indicates the Travenanzes Fm., showing a lateral transition in facies
and a transgressive boundary to the Dolomia Principale. Compiled from Breda and Preto
(2011), after De Zanche et al. (1993), modified.

**Figure 2.** Stratigraphic section at Rifugio Dibona: (**a**) Complete section modified after Breda
and Preto (2011); (**a**) detailed section of uppermost part of the clay-rich interval, showing
sampling locations. (**c**) Outcrop photograph showing the uppermost grey part of the clay-rich
interval with the location of the profile shown in (b).

**Figure 3.** Outcrop images of different types of dolomite intercalated in red and grey clay of
the Travenanzes Fm. at Rifugio Dibona: (**a**) Homogeneous dolomite bed (15 cm thick; 33 m).
(**b**) Upper part: dolomite nodules embedded in red clay, crosscut by green coloured cracks as
part of a calcic vertisol (95 m). (**c**) Laminated dolomite (110-112 m) in grey clay. (**d**) Bed
with gypsum nodules, and cracks filled with gypsum, at 50 m; (**e**) Dolomite-cemented
conglomerate bed at 75 m. (**f**) Laminated bed showing soft sediment deformation (106 m); an
isoclinal synsedimentary fold is indicated by the arrow. (**g**) Laminated dolomite showing
folding of the laminae due to soft sediment deformation (same bed as in f).

**Figure 4.** Photomicrographs of thin sections of dolomites of the Travenanzes Fm.: (**a**)
Rounded mud clasts embedded in dolomicrite matrix. The larger mm-size intraclast in the
upper left side of the image (arrow) consists itself of matrix with darker embedded mudclasts
(sample TZ16-St1; 104 m). (**b**) Mud clasts in dolomicrite matrix. Mudclasts are deformed and



layers of coarser and finer matrix are equally affected by plastic deformation (sample TZ16-
22; 120 m). (**c**, **d**) Pseudomorphs after gypsum in fine-grained dolomudstone (arrows). (**e**)
Oolitic grainstone (sample TZ14-4; 64 m). The cortices consist of microcrystalline dolomite
lacking a radial structure. Some show a concentric structure (arrow). (**f**) Laminated dolomite
showing pseudo-teepee structures (arrow). Vertical cracks are often, but not always,
associated with pseudo-teepees (sample TZ14-10; 107 m). Some coarser grained laminae may
contain microsparite and peloids (P). (**g**) Laminated dolomite showing both plastic and brittle
deformation of laminae. A cm-scale pseudo-teepee occurs in the centre of the image (sample
TZ 16-21; 107 m). (**h**, **i**) Closeup of graded lamina in (g) showing plastic deformation. The
top of the lamina shows an erosion surface with small rip-up clasts (arrow), overlain by a
coarser layer.

**Figure 5.** SEM images of dolomites in backscatter mode: (**a**) Overview showing layer
enriched in celestine inclusions (bright areas) in dolomite (Sample TZ14-9d; 95 m); (**b**)
Celestine inclusion with barite in the centre (same sample as in a); (**c**) Barite crystals in
dolomicrite (sample TZ14-4; 65 m).

**Figure 6.** SEM images of dolomites in backscatter mode showing different types of crystal
shape: (**a**) Spheroidal growth of dolomite (darker areas) in clay layers (brighter areas; sample
TZ14-9d; 95 m). (**b**) Closeup of a. (**c**) Dolomite crystals showing a porous interior domain but
homogeneous syntaxial cement rims (sample TZ14-12; 90 m). (**d**) Similar as in (c; sample
TZ14-9d; 95 m).

**Figure 7.** X-ray diffraction patterns: (**a**) Bulk analyses of homogeneous dolomite (Samples
TZ14-1, TZ14-7, and TZ14-9); main peaks and ordering peaks are labelled with (hkl) indices.
(**b**-**d**) Clay mineral separates of samples TZ14-1, TZ14-7 and TZ14-9, air dried (N), saturated



with ethylene glycol (EG), and heated to 550°C (T); d-values in Å. In the ethylene-glycol
saturated sample TZ14-9 the illite-smectite mixed-layer is best seen. The arrow points at the
expandable (smectite) part of the mixed-layer.

**Figure 8.** (**a**) Carbon/oxygen isotope cross-plot. (**b**) Oxygen isotope values through the
stratigraphic section.

**Figure 9.** Element concentrations in sequentially extracted fractions of bulk dolomite and
clay samples of the Travenanzes Fm.: (**a**) Ca vs. Mg; (**b**) Sr vs. K.

**Figure 10.** Comparison of Sr-isotopes in dolomites of the Travenanzes Fm. with Carnian
seawater curve (Korte et al., 2003). The 2-sigma uncertainties are smaller than the symbol
size.

**Figure 11.** Sr-isotope values from dolomites of different modern and ancient environments:
Travenanzes Fm. in the Dolomites, Southern Alps; Germanic Keuper (Weser Formation and
Arnstadt Formation); Coorong Lagoon; Deep Springs Lake. $^{87}Sr/^{86}Sr$ ratios of modern
seawater are from DePaolo and Ingram (1985).

**TABLES**
**Table 1.** Compilation of sedimentary structures from thin section analysis of dolomites from
the Travenanzes Fm. at the Dibona section.

**Table 2.** Relative abundances and ordering parameters of dolomites from the Travenanzes
Formation. Relative abundances were estimated based on the 104 peak height. The
stoichiometry Mg/(Ca+Mg) was determined from the shift of the 104 peak using the equation





of Lumsden (1979). The structural ordering was calculated from the ratio of the 015 ordering
peak to the 110 peak according to Füchtbauer and Goldschmidt (1966).

**Table 3.** Total inorganic and organic carbon (TIC, TOC) contents of clay samples from the
Travenanzes Formation.

**Table 4.** Carbon and oxygen isotope values of different types of dolomite from the
Travenanzes Formation.

**Table 5.** Element concentrations of leacheates from dolomites and clays used for Sr-isotope
analysis.

**Table 6.** Compiled $^{87}Sr/^{86}Sr$ ratios of sequentially leached dolomites from different locations,
clays and test minerals using different extraction solutions.





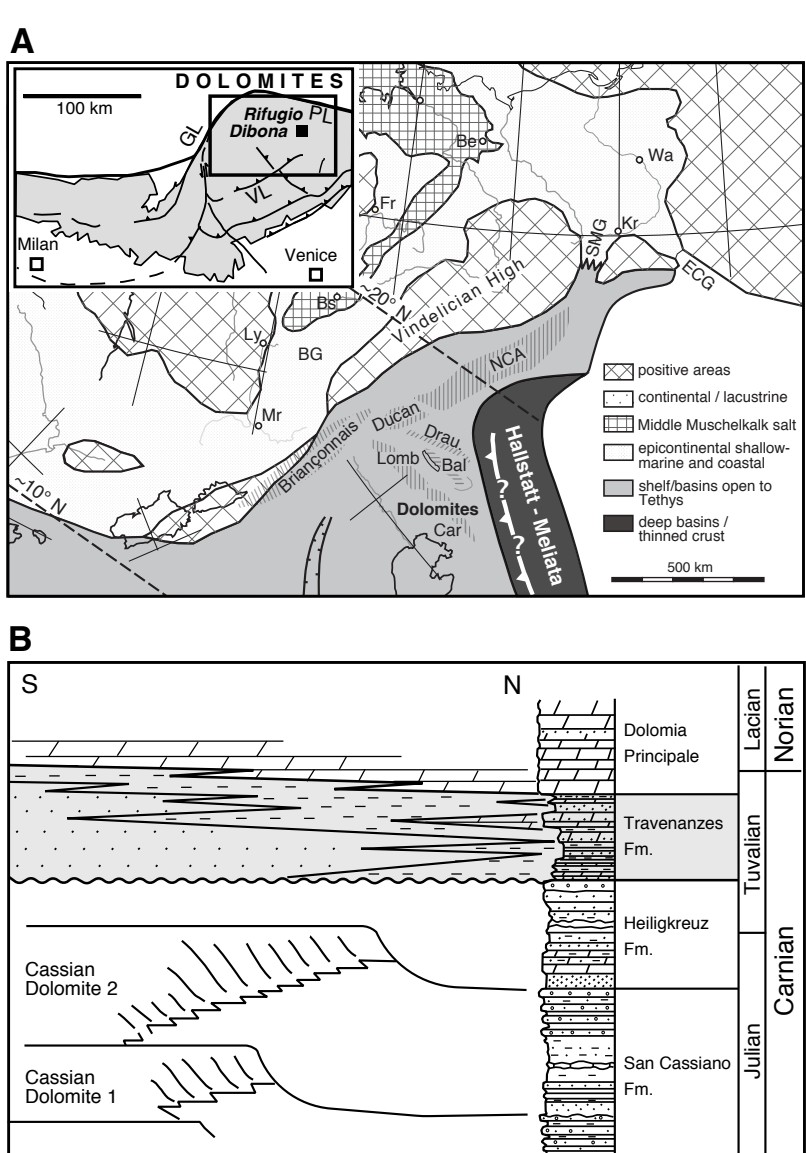

Figure 1

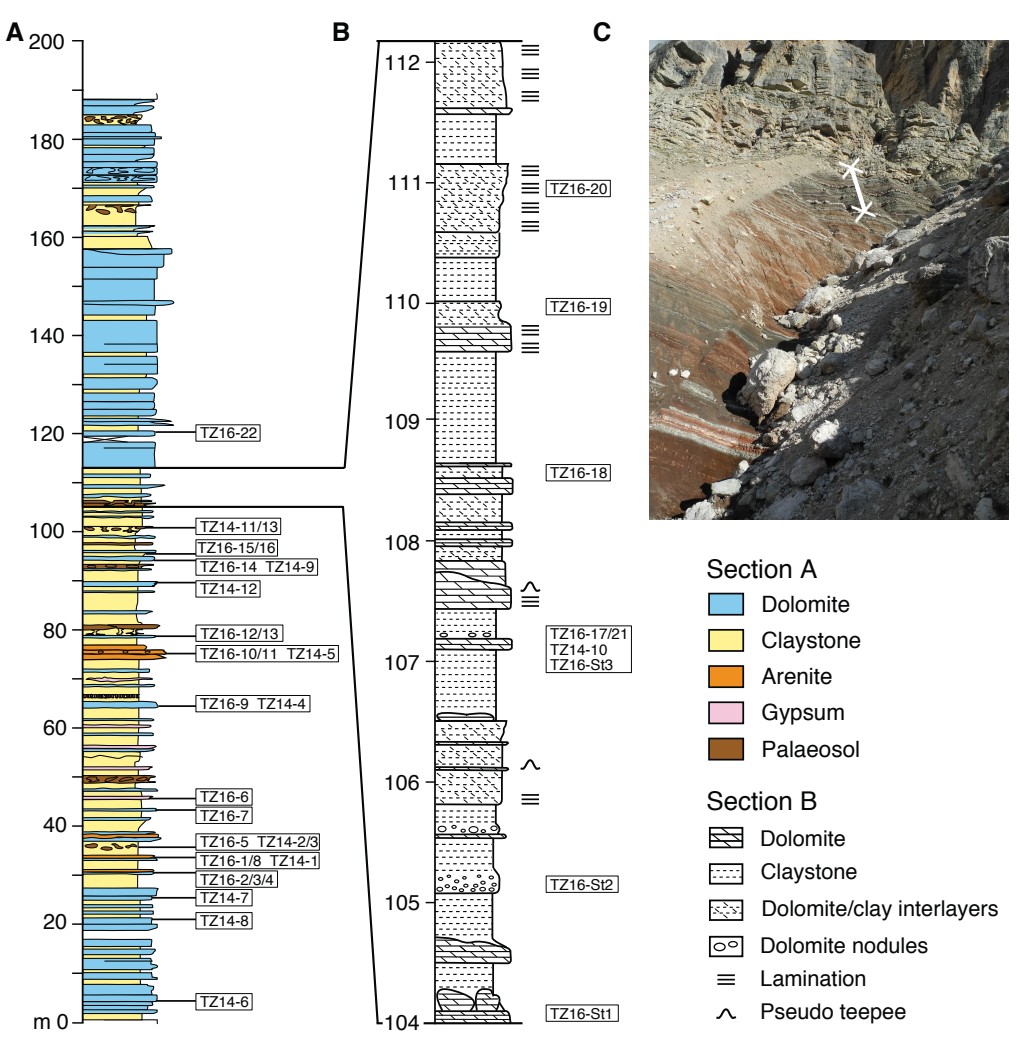

Figure 2




Figure 3





Figure 4



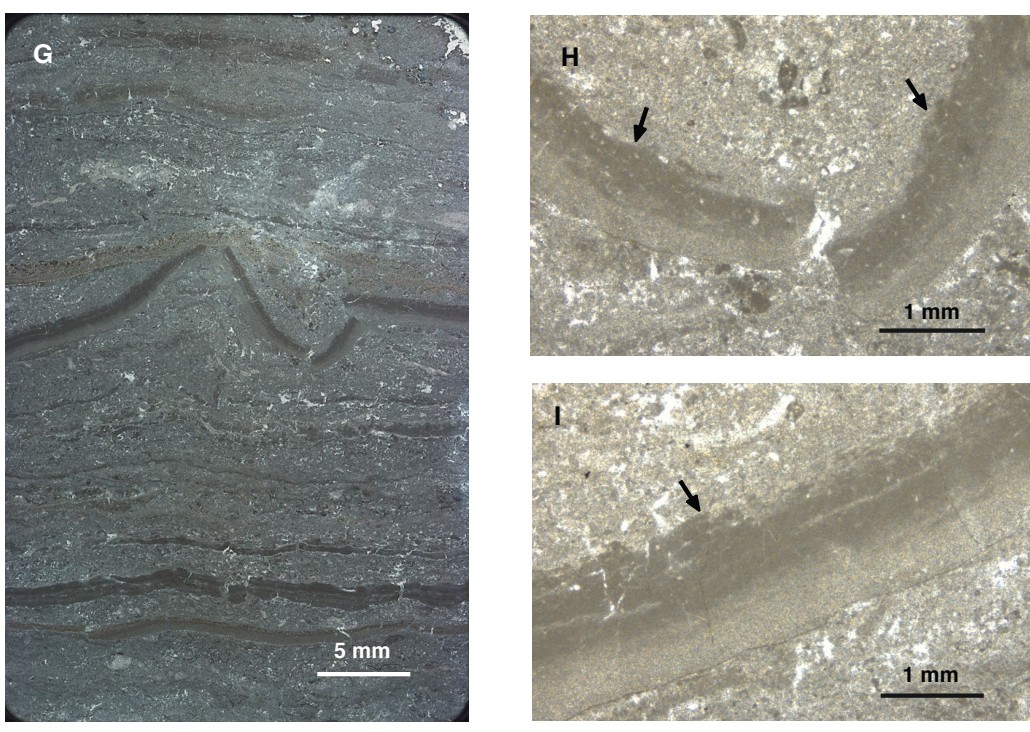

Figure 4 continued





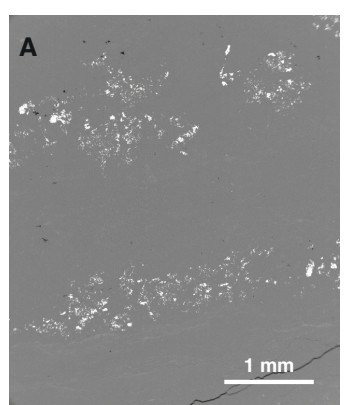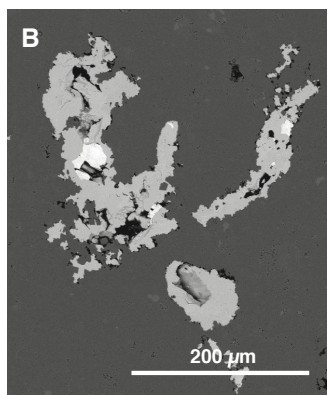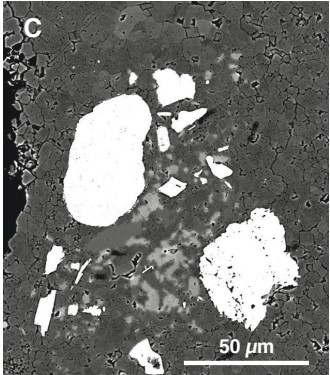

Figure 5





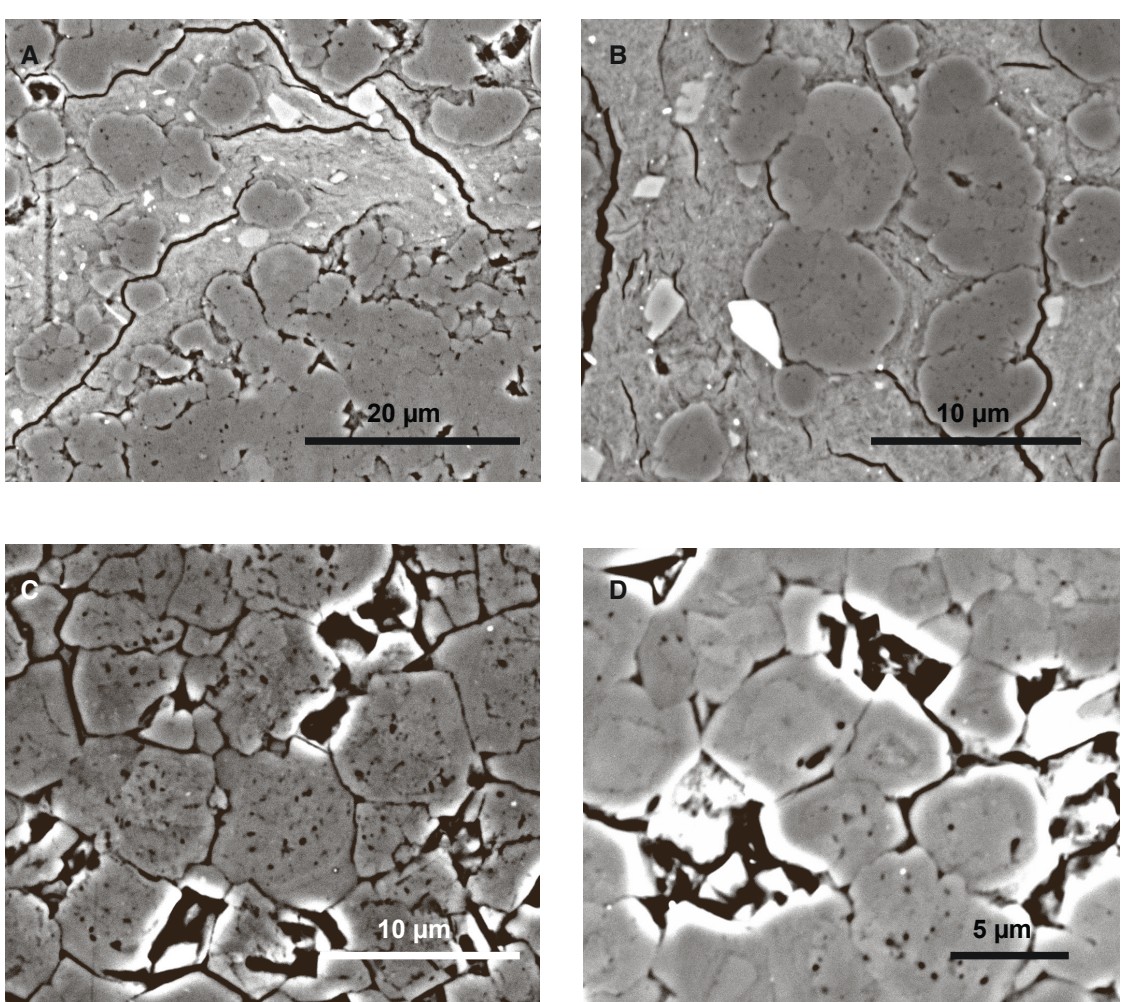

Figure 6

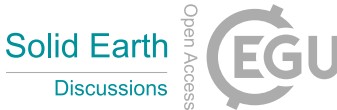

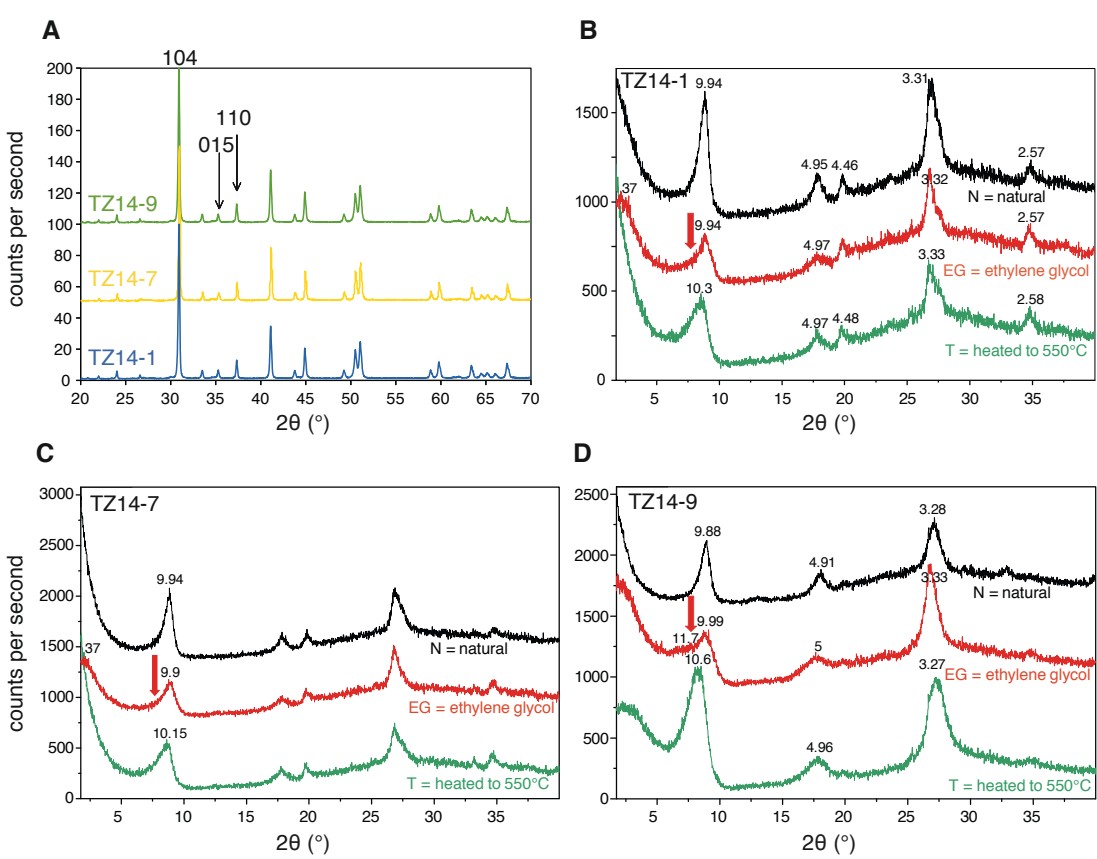

Figure 7




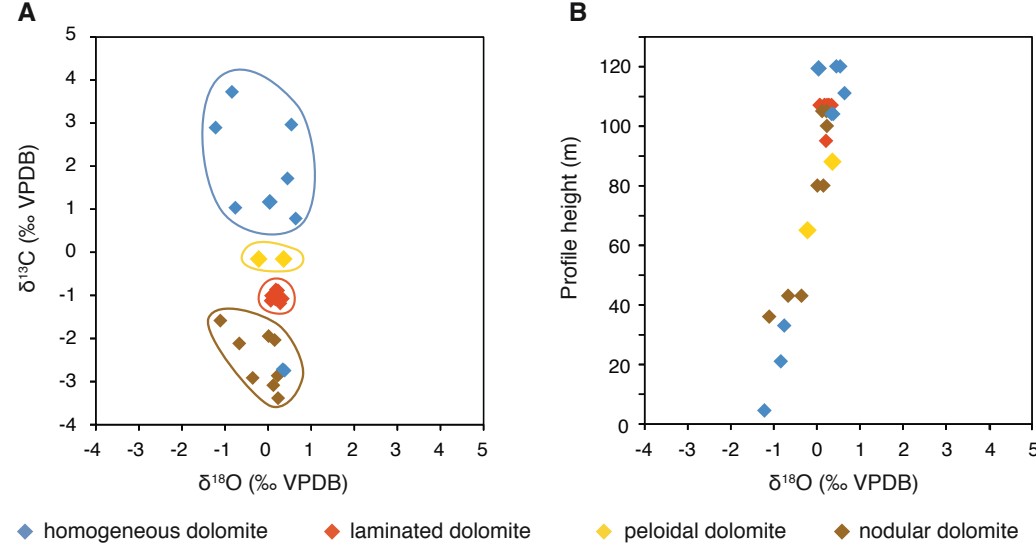

Figure 8



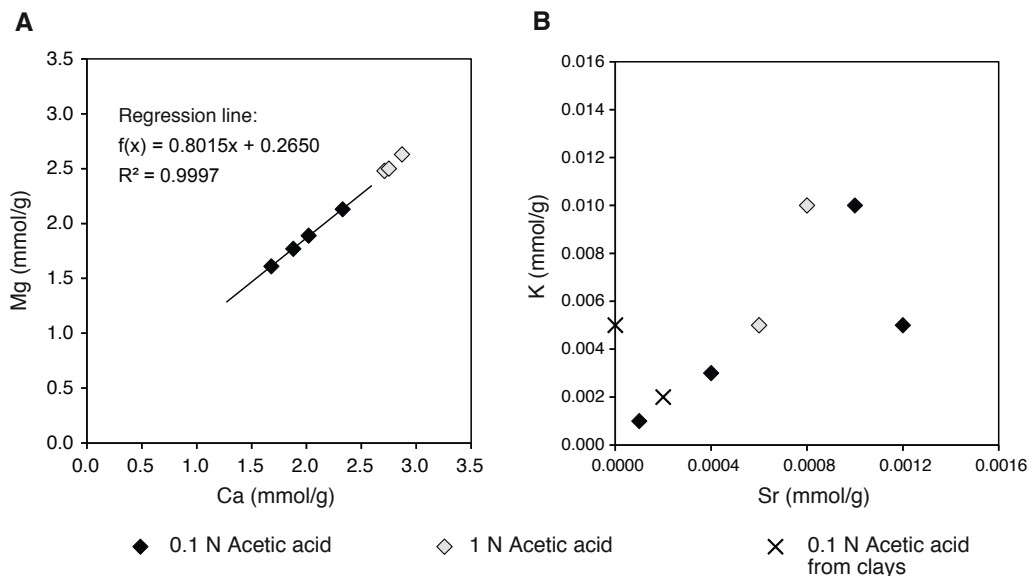

Figure 9



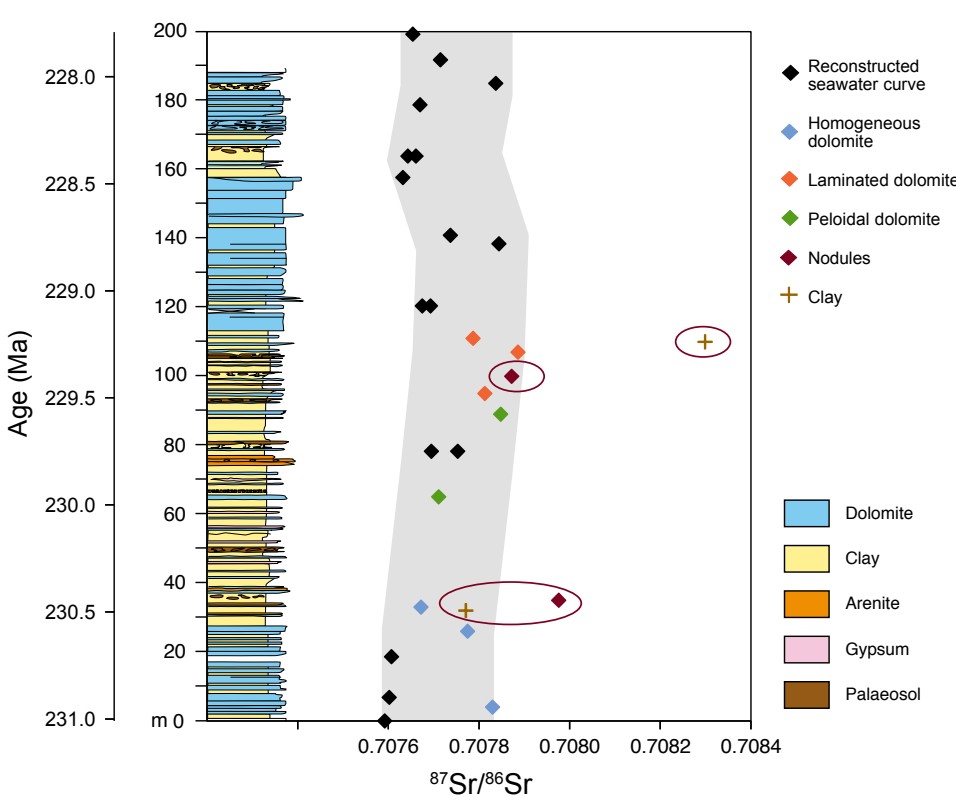

Figure 10





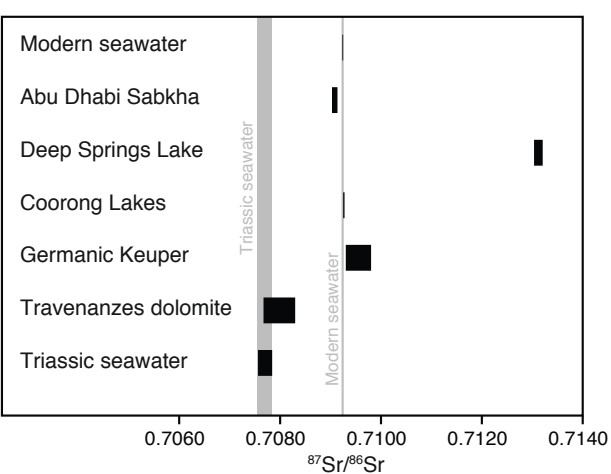

Figure 11



**Table 1**

| Macroscopic Description | Height | Samples 2014 | Samples 2016 | Matrix Aphano-topic | Micro-spar | Cavity-filling | Allochems Undeform. mud clasts* | Deformed mud clasts | Flat pebbles | Packed peloids | Ooids | Quartz clasts | Bioclasts | Sedimentary structures Lamination | Graded Bedding | Pseudo-Teepee | Erosion surfaces | Porosity Fenestral Porosity | Moldic porosity | Deformation Soft sediment deformation | Brittle deformation | SMF** |
|---|---|---|---|---|---|---|---|---|---|---|---|---|---|---|---|---|---|---|---|---|---|---|
| dolomite with pyrite | 120.0 | | TZ16-22 | + | + | + | ++ | + | - | + | - | - | | + | - | - | + | + | + | + | ++ | 25 |
| laminated dolomite/clay | 111.0 | TZ14-11 | TZ16-20 | ++ | + | + | + | ++ | + | ± | - | - | | ++ | + | + | ± | + | - | + | ++ | 25 |
| laminated dolomite | 107.0 | TZ14-10 | TZ16-21/St-3 | + | + | + | + | ++ | + | + | - | - | | ++ | + | + | ++ | + | - | + | + | 25 |
| laminite | 107.0 | | TZ16-i7 | ++ | + | - | + | + | ± | - | - | - | + | ± | ± | + | - | + | ++ | + | + | 25 |
| dolomite nodule | 105.0 | | TZ16-St-2 | ++ | + | - | + | + | - | ± | - | - | | - | - | - | - | - | - | + | + | diag. |
| homogeneous dolomite | 104.0 | | TZ16-St-1 | ++ | + | ± | + | ± | - | ± | - | - | | - | - | - | - | - | - | + | + | 23 |
| nodular dolomite | 100.0 | TZ14-13 | | ++ | + | - | ± | ± | ± | ± | - | - | | - | - | - | - | - | - | ± | + | diag. |
| dolomite nodule | 96.0 | | TZ16-i5 | ++ | - | + | ++ | + | ± | ± | - | - | | - | - | + | - | - | - | ++ | ++ | diag. |
| laminated dolomite | 95.0 | TZ14-9 | | + | + | - | + | + | ± | - | - | - | | ++ | + | + | + | + | - | + | + | 25 |
| porous dolomite | 89.0 | TZ14-12 | | + | ± | + | + | ± | - | - | - | - | Forams Biocl. | - | - | - | - | + | ++ | + | + | 23 |
| nodular bed | 80.0 | | TZ16-13 | + | + | ± | + | ± | - | - | - | - | (+) | - | - | - | - | ± | + | ++ | - | diag. |
| palaeosol, nodule | 79.0 | | TZ16-12 | + | ± | - | ++ | ± | ++ | - | - | - | | - | - | - | - | ± | + | + | ++ | diag. |
| sandy dolomite (congl.) | 75.0 | TZ14-5 | TZ16-11 | ++ | ± | ± | ++ | - | - | - | - | + | Ostracods (+) | - | - | - | - | - | - | - | - | 23 |
| sandstone | 74.0 | | TZ16-10 | + | ± | - | ++ | - | - | - | - | + | | - | - | - | - | - | - | - | - | 25 |
| brittle porous dolomite | 65.0 | TZ14-4 | | + | - | + | ± | - | - | ± | + | - | Forams + | (-) | + | - | - | + | + | + | + | 15 |
| porous dolomite | 43.0 | | TZ16-7 | ++ | ± | + | ± | ± | - | - | - | - | Ostracods ± | - | - | - | - | ± | + | ++ | + | 23 |
| nodules of dolomite | 35.0 | TZ14-3 | | ++ | + | - | - | ± | - | - | - | - | | - | - | - | - | - | - | + | + | diag. |
| dolomite nodule | 35.0 | TZ14-2 | | + | ± | - | + | ± | - | - | - | - | | - | - | - | - | - | - | + | + | diag. |
| arenite | 33.5 | | TZ16-8 | + | ± | - | + | - | - | ++ | - | ++ | | - | + | - | - | - | - | + | + | Sst. |
| homogeneous dolomite | 33.0 | TZ14-1 | | ++ | ± | - | ± | + | - | ± | - | ± | | ± | ± | - | - | - | - | + | + | 23 |
| laminated dolomite | 31.5 | | TZ16-2 | + | - | - | + | + | - | - | - | ++ | | + | + | - | + | + | - | + | ± | Sst. |
| palaeosol, dolomit | 30.1 | | TZ16-3 | ++ | ± | ± | + | ± | - | - | - | - | | - | - | - | + | - | - | + | + | diag. |
| red mottled dolomite | 26.0 | TZ14-7 | | - | - | - | ± | ± | - | - | - | - | | ± | - | - | ± | - | - | ± | ± | 23 |
| dolomite with clay | 21.0 | TZ14-8 | | ++ | ± | - | + | ± | - | - | - | - | | + | - | - | - | - | - | - | - | 23 |
| homogeneous dolomite | 4.0 | TZ14-6 | | ++ | + | ± | + | ± | + | ± | - | - | Forams + | + | + | + | + | - | + | + | ± | 25 |

- not present    ± rare    + common    ++ very abundant    (+) putative

\* Needs to be further subdivided into peloids, intraclasts, flat pebbles and clast of brittle deformation

\*\* Nodules are most likely diagenetic and can thus not be associated to a microfacies

Legend: Homogeneous dolomite | Laminated dolomite | Nodular dolomite | Oolithic dolomite | Sandstone





**Table 2**

| Sample | Depth (m) | d(A°) | Ca/(Ca+Mg) (%) | 015/110 |
|--------|-----------|---------|----------------|---------|
| TZ14-1 | 33.0 | 2.88944 | 51.1 | 0.44 |
| TZ14-7 | 43.0 | 2.88871 | 50.9 | 0.41 |
| TZ14-9 | 95.0 | 2.88633 | 50.1 | 0.46 |



**Table 3**

| Sample | TC (wt%) | TOC (wt%) | TIC (wt%) |
|---|---|---|---|
| TZ16-1 | 0.06 | 0.05 | 0.02 |
| TZ16-19B | 0.12 | 0.11 | 0.02 |
| TZ16-5 | 0.16 | 0.05 | 0.12 |
| TZ16-19A | 0.34 | 0.10 | 0.25 |
| TZ16-14 | 0.42 | 0.16 | 0.27 |
| TZ16-18 | 0.50 | 0.07 | 0.43 |
| TZ16-16 | 0.51 | 0.05 | 0.46 |




**Table 4**

| Sample | Depth (m) | $\delta^{13}$C (‰ VPDB) | $\delta^{18}$O (‰ VPDB) | Type | Description |
|---|---|---|---|---|---|
| TZ14-1 | 33 | 1.04 | -0.76 | homogeneous | with siliciclastis |
| TZ14-3 | 35 | -1.58 | -1.11 | nodule | with barite |
| TZ14-4 | 65 | -0.15 | -0.22 | peloidal | with apatite |
| TZ14-6 | 4 | 2.90 | -1.22 | homogeneous | with siderite and pyrite |
| TZ14-8 | 21 | 3.73 | -0.84 | homogeneous | with clay, apatite and Fe-oxide |
| TZ14-9 | 95 | -1.01 | 0.21 | laminated | with celestine and barite |
| TZ14-10b | 107 | -1.05 | 0.26 | laminated | with apatite and pyrite |
| TZ14-11 | 111 | 0.79 | 0.64 | homogeneous | homog. Lamina with clay and pyrite |
| TZ14-12 | 89 | -0.15 | 0.36 | peloidal | with megalodont and Ti-oxides |
| TZ14-13 | 100 | -3.38 | 0.23 | nodule | palaeosol with Fe-oxide |
| | | | | | |
| TZ16-St1 | 104 | -2.74 | 0.38 | homogeneous | mud clast top |
| | 104 | -2.71 | 0.34 | homogeneous | matrix |
| TZ16-St2 | 105 | -3.08 | 0.12 | nodule | matrix top |
| | 105 | -2.86 | 0.22 | nodule | matrix bottom |
| TZ16-7 | 43 | -2.91 | -0.36 | *Rauhwacke* | matrix top |
| | 43 | -2.11 | -0.67 | *Rauhwacke* | matrix bottom |
| TZ16-13 | 80 | -2.03 | 0.15 | nodule | matrix top |
| | 80 | -1.94 | 0.01 | nodule | matrix bottom |
| TZ16-21 | 107 | -0.86 | 0.18 | laminated | graded lamina |
| | 107 | -1.09 | 0.17 | laminated | ligh lamina |
| | 107 | -0.88 | 0.23 | laminated | dark lamina |
| TZ16-22 | 120 | 1.72 | 0.45 | homogeneous | mud clast |
| | 120 | 1.22 | 0.15 | homogeneous | lamination |
| | 120 | 2.97 | 0.54 | homogeneous | homogeneous part |
| TZ16-St3 | 107 | -1.07 | 0.34 | laminated | dark layer, lense |
| | 107 | -1.00 | 0.07 | laminated | dark layer bottom |
| | 107 | -1.18 | 0.28 | laminated | dark layer top |
| | 107 | -1.12 | 0.06 | laminated | light layer top |





**Table 5**

| Sample | Element | 0.1 N acetic acid fraction | | | | 1 N acetic acid fraction | | | 1 N HCl fraction | |
|---|---|---|---|---|---|---|---|---|---|---|
| | | µmol/g | µmol/g | µmol/g | µmol/g | µmol/g | µmol/g | µmol/g | µmol/g | µmol/g |
| *Bulk dolomite samples* | | | | | | | | | | |
| | | TZ14-1 | TZ14-7 | TZ14-9 | crystal | TZ14-1 | TZ14-7 | TZ14-9 | | |
| | | 0.098 g | 0.127 g | 0.099 g | 0.094 g | 0.098 g | 0.127 g | 0.099 g | | |
| | Al | 6.58 | 3.17 | 11.57 | 3.04 | 4.51 | 8.17 | 5.97 | | |
| | Ca (mmol/g) | 1.68 | 2.33 | 1.88 | 2.50 | 2.87 | 2.71 | 2.75 | | |
| | Fe | 4.97 | 3.53 | 10.67 | 34.27 | 2.04 | 9.15 | 5.02 | | |
| | K | 3.32 | 9.71 | 5.26 | 0.93 | 10.31 | 4.65 | 13.51 | | |
| | Mg (mmol/g) | 1.61 | 2.13 | 1.77 | 2.34 | 2.64 | 2.48 | 2.50 | | |
| | Mn | 5.96 | 3.57 | 7.68 | 15.24 | 10.84 | 3.72 | 10.67 | | |
| | Na | 12.78 | 18.98 | 17.32 | 1.85 | 17.35 | 20.12 | 23.30 | | |
| | P | 1.50 | n.d. | 0.98 | n.d. | 0.20 | 1.45 | n.d. | | |
| | Ti | n.d. | n.d. | n.d. | n.d. | n.d. | n.d. | n.d. | | |
| | Ba | 0.50 | 0.03 | 0.48 | n.d. | 1.75 | 0.02 | 1.03 | | |
| | Sr | 0.38 | 1.00 | 1.16 | 0.13 | 0.79 | 0.57 | 34.91 | | |
| | Rb | n.d. | n.d. | n.d. | n.d. | n.d. | n.d. | n.d. | | |
| *Clay samples* | | | | | | | | | | |
| | | TZ16-1 | TZ16-19B | | | TZ16-1 | TZ16-19B | | TZ16-1 | TZ16-19B |
| | | 0.038 g | 0.030 g | | | 0.038 g | 0.030 g | | 0.038 g | 0.030 g |
| | Al | 2.18 | 4.52 | | | 1.54 | 4.01 | | 39.86 | 33.62 |
| | Ca | 19.14 | 11.62 | | | 8.48 | 4.34 | | 0.71 | 0.60 |
| | Fe | 0.72 | 1.79 | | | 0.83 | 2.25 | | 75.59 | 11.56 |
| | K | 4.97 | 9.02 | | | 2.77 | 3.69 | | 11.69 | 12.61 |
| | Mg | 8.05 | 13.76 | | | 4.46 | 6.07 | | 24.62 | 18.75 |
| | Mn | n.d. | n.d. | | | n.d. | n.d. | | n.d. | n.d. |
| | Na | 0.355 | 0.470 | | | 0.305 | 0.389 | | 0.531 | 0.828 |
| | P | 6.89 | 1.08 | | | 0.67 | n.d. | | n.d. | n.d. |
| | Ti | n.d. | n.d. | | | n.d. | n.d. | | 1.305 | 0.194 |
| | Ba | n.d. | n.d. | | | n.d. | n.d. | | 0.022 | n.d. |
| | Sr | 0.417 | 0.047 | | | 0.187 | 0.018 | | 0.017 | 0.005 |
| | Rb | n.d. | n.d. | | | n.d. | n.d. | | n.d. | n.d. |

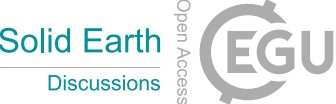

**Table 6. Sr-isotopes**

| Sample | Section (m) | Description | Seq. extr. | Weight (mg) | Reagent | Amount (ml) | Extr. T (°C) | Extr. time | Shaker y/n | Washing (before step) | Run no. | $^{87}Sr/^{86}Sr$ | $2\sigma (10^6)$ | Aliquot for conc. |
|---|---|---|---|---|---|---|---|---|---|---|---|---|---|---|
| NBS987 | | Standard solution (500 ppm) | | 500 ng | | | | | | | (n = 40) | 0.710272 | 4 | |
| NBS988 | | Standard solution (500 ppm) | | 500 ng | | | | | | | (n = 9) | 0.710268 | 6 | |
| **Test minerals** | | | | | | | | | | | | | | |
| *Series 1 (sequential extraction)* | | | | | | | | | | | | | | |
| Celestine | | | | 2.34 | 1M NaCl | 2 ml | 20 | 12 h | n | 1M NaCl | 6052 | 0.708037 | 5 | |
| Barite | | | | 25.09 | 0.1N AcOH | 2 ml | 20 | 12 h | n | 3M NaCl, 1M KCl, H₂O, 0.1N AcOH | 6109 | 0.708887 | 9 | |
| Dolomite | | | | 9.88 | 0.1N AcOH | 2 ml | 20 | 12 h | n | 3M NaCl, 1M KCl, H₂O, 0.1N AcOH | 6110 | 0.709942 | 11 | |
| Mixture | | Barite 4.5 mg; Celestine 8.91 mg; Dolomite 35.9 mg | seq. | 49.31 | 1M NaCl | 2 ml | 20 | 12 h | n | 1M NaCl | 6053 | 0.708038 | 3 | |
| Mixture | | | seq. | - | 0.1N AcOH | 2 ml | 20 | 12 h | n | 3M NaCl, 1M KCl, H₂O, 0.1N AcOH | 6108 | 0.709501 | 40 | |
| *Series 2* | | | | | | | | | | | | | | |
| Celestine | | | | 2.22 | 1M NaCl | 2 ml | 20 | 2 h | n | - | 6121 | 0.708045 | 4 | |
| Celestine | | | | 4.60 | 0.1N AcOH | 2 ml | 20 | 4 h | n | 12h 1M NaCl | 6132 | 0.708047 | 3 | |
| Barite | | | | 36.94 | 6N HCl | 2 ml | 40 | 12 h | n | - | 6152 | 0.707610 | 5 | |
| Barite | | | seq | - | 6N HCl | 2 ml | 40 | 12 h | n | - | 6155 | 0.707564 | 6 | |
| Dolomite | | | | 17.37 | 0.1N AcOH | 5 ml | 40 | 12 h | n | - | 6068 | 0.710831 | 7 | |
| Dolomite | | Replicate | | 3.41 | 0.1N AcOH | 2 ml | 40 | 12 h | n | - | 6114 | 0.710557 | 11 | |
| **Travenanzes Fm.** | | | | | | | | | | | | | | |
| *Bulk samples sequential extractions, Series 1* | | | | | | | | | | | | | | |
| TZ14-1 | 33 m | homogeneous dolomite | seq. | 12.35 | 1M NaCl | 2 ml | 20 | 12 h | n | - | 6112 | 0.708125 | 12 | |
| TZ14-1 | 33 m | homogeneous dolomite | seq. | - | 0.1N AcOH | 3 × 2 ml | 20 | 4h, 12h, 4h | y | 1M NaCl, H₂O, 3.3M KCl. H₂O | 6169 | 0.707666 | 4 | |
| TZ14-1 | 33 m | homogeneous dolomite | seq. | - | 0.1N AcOH | 2 ml | 20 | 36 h | y | - | 6173 | 0.715417 | 250 | |
| TZ14-9 | 95 m | laminated dolomite | seq. | 13.50 | 1M NaCl | 2 ml | 20 | 12 h | n | - | 6113 | 0.707880 | 4 | |
| TZ14-9 | 95 m | laminated dolomite | seq. | - | 0.1N AcOH | 3 × 2 ml | 20 | 4h, 12h, 4h | y | 1M NaCl, H₂O, 3.3M KCl. H₂O | 6171 | 0.707817 | 5 | |
| TZ14-9 | 95 m | laminated dolomite | seq. | - | 0.1N AcOH | 2 ml | 20 | 36 h | y | - | 6174 | 0.719226 | 455 | |
| Mixture | | Residue from test mineral series 1 | seq. | - | 0.1N AcOH | 3 × 2 ml | 20 | 4h, 12h, 4h | y | 1M NaCl, H₂O, 3.3M KCl. H₂O | 6172 | 0.709812 | 5 | |
| Mixture | | Residue from test mineral series 1 | seq. | - | 0.1N AcOH | 2 ml | 20 | 36 h | y | - | 6176 | 0.709900 | 4 | |
| TZ14-1 | 33 m | homogeneous dolomite | | 42.76 | 0.1N AcOH | 2 ml | 20 | 4 h | n | - | 6130 | 0.707894 | 4 | |
| TZ14-9 | 95 m | laminated dolomite | | 17.69 | 0.1N AcOH | 2 ml | 20 | 4 h | n | - | 6131 | 0.707872 | 5 | |
| *Bulk samples sequential extractions, Series 2* | | | | | | | | | | | | | | |
| TZ14-1 | 33 m | homogeneous dolomite | | 93.91 | 1M NaCl | 10 ml | 20 | 12 h | y | - | 6182 | 0.708096 | 5 | |
| TZ14-1 | 33 m | homogeneous dolomite | seq. | 98.28 | 0.1N AcOH | 10 ml | 20 | 12 h | y | - | 6183 | 0.707812 | 4 | yes |
| TZ14-1 | 33 m | homogeneous dolomite | seq. | - | 1N AcOH | 10 ml | 20 | 12 h | y | - | 6205 | 0.707670 | 5 | yes |
| TZ14-1 | 33 m | homogeneous dolomite | | 50.00 | 6N HCl | 5 ml | 20 | 12 h | n | 10h 1N CH₃COOH | 6445 | 0.710403 | 6 | |
| TZ14-7 | 26 m | mottled dolomite | | 90.64 | 1M NaCl | 10 ml | 20 | 12 h | y | - | 6179 | 0.707883 | 4 | |
| TZ14-7 | 26 m | mottled dolomite | seq. | 127.52 | 0.1N AcOH | 10 ml | 20 | 12 h | y | - | 6178 | 0.707801 | 4 | yes |
| TZ14-7 | 26 m | mottled dolomite | seq. | - | 1N AcOH | 10 ml | 20 | 12 h | y | - | 6207 | 0.707719 | 4 | yes |
| TZ14-7 | 26 m | mottled dolomite | | 50.00 | 6N HCl | 5 ml | 20 | 12 h | n | 10h 1N CH₃COOH | 6449 | 0.730453 | 5 | |
| TZ14-9 | 95 m | laminated dolomite | | 97.82 | 1M NaCl | 10 ml | 20 | 12 h | y | - | 6187 | 0.707869 | 3 | |
| TZ14-9 | 95 m | laminated dolomite | seq. | 98.76 | 0.1N AcOH | 10 ml | 20 | 12 h | y | - | 6185 | 0.707862 | 3 | yes |
| TZ14-9 | 95 m | laminated dolomite | seq. | - | 1N AcOH | 10 ml | 20 | 12 h | y | - | 6206 | 0.707813 | 3 | yes |
| TZ14-9 | 95 m | laminated dolomite | | 50.00 | 6N HCl | 5 ml | 20 | 12 h | n | 10h 1N CH₃COOH | 6447 | 0.708464 | 4 | |
| Dolomite (single crystal) | | control | | 116.65 | 1M NaCl | 10 ml | 20 | 12 h | y | - | 6184 | 0.708401 | 40 | |
| Dolomite (single crystal) | | control | seq. | 94.12 | 0.1N AcOH | 10 ml | 20 | 12 h | y | - | 6180 | 0.707735 | 6 | yes |
| Dolomite (single crystal) | | control | seq. | - | 1N AcOH | 10 ml | 20 | 12 h | y | - | 6208 | 0.707666 | 6 | yes |
| *Micro-drill samples* | | | | | | | | | | | | | | |
| TZ14-3 | 35 m | dolomite nodule | | - | 0.1N AcOH | 2 ml | 20 | 24 h | n | H₂O, 5min 0.1N AcOH | 6548 | 0.707976 | 4 | |
| TZ14-4 | 65 m | peloidal dolomite | | - | 0.1N AcOH | 2 ml | 20 | 24 h | n | H₂O, 5min 0.1N AcOH | 6549 | 0.707711 | 4 | |
| TZ14-6 | 4 m | homogeneous dolomite | | - | 0.1N AcOH | 2 ml | 20 | 24 h | n | H₂O, 5min 0.1N AcOH | 6550 | 0.707830 | 4 | |
| TZ14-8 | 21 m | dolomite with clay | | - | 0.1N AcOH | 2 ml | 20 | 24 h | n | H₂O, 5min 0.1N AcOH | 6551 | 0.707821 | 4 | |
| TZ14-10b | 107 m | laminated dolomite | | - | 0.1N AcOH | 2 ml | 20 | 24 h | n | H₂O, 5min 0.1N AcOH | 6554 | 0.707886 | 4 | |
| TZ14-11 | 111 m | laminated dolomite | | - | 0.1N AcOH | 2 ml | 20 | 24 h | n | H₂O, 5min 0.1N AcOH | 6553 | 0.707787 | 4 | |
| TZ14-12 | 89 m | peloidal dolomite | | - | 0.1N AcOH | 2 ml | 20 | 24 h | n | H₂O, 5min 0.1N AcOH | 6555 | 0.707848 | 4 | |
| TZ14-13 | 100 m | dolomite with palaeosol | | - | 0.1N AcOH | 2 ml | 20 | 24 h | n | H₂O, 5min 0.1N AcOH | 6556 | 0.707872 | 4 | |
| *Micro-drilled sequential extractions* | | | | | | | | | | | | | | |
| TZ14-1 | 33 m | homogeneous dolomite | seq. | - | 0.1N AcOH | 2 ml | 20 | 24 h | n | H₂O, 5min 0.1N AcOH | 6411 | 0.707672 | 3 | |
| TZ14-1 | 33 m | homogeneous dolomite | seq. | - | 1N AcOH | | 20 | 24 h | n | H₂O, 5min 0.1N AcOH | 6448 | 0.708300 | 23 | |
| TZ14-7 | 26 m | mottled dolomite | seq. | - | 0.1N AcOH | 2 ml | 20 | 24 h | n | H₂O, 5min 0.1N AcOH | 6479 | 0.707775 | 6 | |
| TZ14-7 | 26 m | mottled dolomite | seq. | - | 1N AcOH | | 20 | 24 h | n | H₂O, 5min 0.1N AcOH | 6444 | 0.708502 | 22 | |
| TZ14-7 | 26 m | clay layer | seq. | - | 0.1N AcOH | | 20 | 24 h | n | H₂O, 5min 0.1N AcOH | 6410 | 0.707742 | 4 | |
| TZ14-7 | 26 m | clay layer | seq. | - | 1N AcOH | | 20 | 24 h | n | H₂O, 5min 0.1N AcOH | 6446 | 0.708467 | 2 | |
| TZ14-9 | 95 m | laminated dolomite | seq. | - | 0.1N AcOH | 2 ml | 20 | 24 h | n | H₂O, 5min 0.1N AcOH | 6412 | 0.707813 | 149 | |
| TZ14-9 | 95 m | laminated dolomite | seq. | - | 1N AcOH | | 20 | 24 h | n | H₂O, 5min 0.1N AcOH | 6443 | 0.708281 | 56 | |
| *Clay samples* | | | | | | | | | | | | | | |
| TZ16-1 | 32 m | red (green) clay | seq. | 38.30 | 0.1N AcOH | 2 ml | 20 | 12 h | n | 2h 0.1N AcOH | 6557 | 0.707771 | 4 | yes |
| TZ16-1 | 32 m | red (green) clay | seq. | - | 1N AcOH | 2 ml | 20 | 12 h | n | 2h 1N AcOH | 6558 | 0.707768 | 4 | yes |
| TZ16-1 | 32 m | red (green) clay | seq. | - | 6N HCl | 2 ml | 20 | 12 h | n | 4h 6N HCl | 6559 | 0.722998 | 18 | yes |
| TZ16-19B | 110 m | dark grey clay | seq. | 29.74 | 0.1N AcOH | 2 ml | 20 | 12 h | n | 2h 0.1N AcOH | 6560 | 0.708299 | 8 | yes |
| TZ16-19B | 110 m | dark grey clay | seq. | - | 1N AcOH | 2 ml | 20 | 12 h | n | 2h 1N AcOH | 6561 | 0.708582 | 8 | yes |
| TZ16-19B | 110 m | dark grey clay | seq. | - | 6N HCl | 2 ml | 20 | 12 h | n | 4h 6N HCl | 6552 | 0.733910 | 24 | yes |
| **Germanic Keuper** | | | | | | | | | | | | | | |
| Lehr | | micro-drilled | | - | 0.1N AcOH | 2 ml | 20 | 24 h | n | H₂O, 5min 0.1N AcOH | 6545 | 0.709303 | 4 | |
| Keu 1-2 B | | micro-drilled | | - | 0.1N AcOH | 2 ml | 20 | 24 h | n | H₂O, 5min 0.1N AcOH | 6546 | 0.709805 | 6 | |
| **Deep Springs Lake** | | | | | | | | | | | | | | |
| DS11-3, 16.5 | 16.5 cm | dried mud | seq. | 91.17 | 0.1N AcOH | 10 ml | 20 | 5 min | n | H₂O | 6363 | 0.713207 | 4 | |
| DS11-3, 16.5 | 16.5 cm | dried mud | seq. | - | 0.1N AcOH | 10 ml | 20 | 10 h | n | H₂O | 6363 | 0.713086 | 4 | |
| DS11-3, 52.5 | 52.5 cm | dried mud | | 62.19 | 0.1N AcOH | 10 ml | 20 | 10 h | n | H₂O | 6343 | 0.713035 | 4 | |
| **Milne Lake (Coorong)** | | | | | | | | | | | | | | |
| CM01-3 | 30 cm | dried mud, ground | seq. | 47.94 | 0.1N AcOH | 10 ml | 20 | 5 min | n | H₂O | 6340 | 0.709251 | 4 | |
| CM01-3 | 30 cm | dried mud, ground | seq. | - | 0.1N AcOH | 10 ml | 20 | 10 h | n | H₂O | 6340 | 0.709275 | 3 | |
| CM01-9 | 90 cm | dried mud, ground | | 52.47 | 0.1N AcOH | 10 ml | 20 | 10 h | n | H₂O | 6341 | 0.709272 | 4 | |