# Peer review of "Precipitation of dolomite from seawater on a Carnian coastal plain (Dolomites, northern"

_Solid Earth, 2019_

## Referee Comment (RC1) · Chris Romanek (Referee) · 15 Mar 2019

I reviewed this manuscript previously for another journal, and while I recommended publication with minor corrections, the editor's decision was 'Reject with Referral Offer'. After closely examining the present and previous manuscript line by line, I conclude there are no substantive changes to the original manuscript other than: 1) the rear-rangement of some material in the introduction section, 2) the deletion or modification of some sentences that did not provide additional insight, 3) the replacement of the word "humid" for "wet", and 4) the addition of a final paragraph just before the conclusions section. As such, my previous review appears relevant so it is posted below.

"This manuscript describes a petrographic and geochemical investigation of dolomitic rocks collected from the Travenanzes formation in northeastern Italy, and dolomite samples collected from the Weser and Arnstadt formations (Germanic Keuper unit), the Coorong Lagoon (Australia) and Deep Springs Lake, CA. Thin sections were prepared from hand samples for petrographic and SEM analysis, and bulk samples and mineral separates were processed for elemental and isotopic (C, O, Sr) analysis. The goal of the study was to '...gain better insight into the conditions and processes of dolomite formation'. Based on the data collected, the investigators propose that the dolomites of the Travenanzes formation were deposited in ephemeral lakes in an extended alluvial plain or dryland river system that was episodically inundated by seawater, and they conclude there is no known modern analogue to this depositional environment.

This manuscript was very well written and it was a pleasure to read. In fact, there were very few places where improvements could be suggested. The summary information provided in the introduction was appropriate and insightful, and the methodology and analytical procedures were explained in a straightforward way. The investigators presented convincing petrographic and geochemical evidence that supports their interpretation for the depositional environment of the Travenanzes dolomites. The Sr isotope data are remarkably consistent with Triassic seawater throughout the length of the section and they only show hints of a continental signature with the most aggressive leaching procedures. The stable O-isotope data are consistent with a marine signature and the C-isotopes demonstrate the incorporation of oxidized organic matter in texturally distinct samples. Overall, the data appear to be straightforward and easy to interpret.

The sequential extraction work (e.g., Table 6) for the Sr-isotope work could be presented better so the reader can understand why various procedures and reagents (e.g., NaCl, AcOH, HCl) were being used. The TIC/TOC results could be integrated in more substantial ways, e.g., perhaps TOC could be related to the development of

dolomite nodules during microbial sulfate reduction."

Several aspects of the present manuscript, that were identified in my previous review, still remain: 1) a general lack of engagement with the bulk elemental data (i.e., Table 5), 2) although 39 samples were collected in this study, it appears only a handful of these are presented in the manuscript for analysis, and 3) the manuscript does not substantially engage the potential for microbial origins although the subject is broached in general ways.

Nevertheless, and as stated in my previous review, ". . . my overall impression is that this manuscript provides a plausible interpretation for the origin of Travenanzes dolomites. This contribution provides an incremental step, albeit a small one, in our general understanding of dolomite formation and more specifically dolomite formation along the Tethyan margin and I feel it is acceptable for publication. . .", although slight modifications are warranted that can be supervised by the associate editor.

---

## Referee Comment (RC2) · Anonymous Referee #2 · 10 Apr 2019

I have read through the paper by Reider and others entitled: "Precipitation of dolomite from seawater on a Carnian coastal 1 plain (Dolomites, northern Italy): evidence from carbonate petrography and Sr-isotopes". I find the paper to be an interesting contribution to our understanding of the processes that led to the formation of primary dolomite in the sedimentary rock record and the authors use some innovative methods to prove a primary origin for the dolomite. However, my main criticism about the paper is that it is too long in its current state, and should be shortened. Examples of text that needs to be edited or cut out altogether include: • The authors spend approximately 4 pages describing their methods, which could be cut down to at least half that length by refer-ring to similar methods in other papers and describing their methods in less detail. • Related to the previous example, the paper contains data (e.g., %TOC, some of the elemental data, etc.) that seems unnecessary to the overall study. I would recommend that the authors carefully go through their manuscript and remove any data that is not deemed essential to their manuscript. • The inclusion of 6 tables seems excessive. The number of tables should be reduced to one or two, with the extra tables relegated to a "Supplemental Materials" section. • It is not clear to me why the authors include analyses of the Germanic Keuper dolomites (lines 359-371) in this paper. • I am not an expert on Sr geochemistry, but it's not clear to me why the authors spend so much time discussing sequential extractions (lines 425-472). It seems to me that this text could be reduced. • Lines 622-638 also seems unnecessary to the paper, as the authors explain one anomalous value from one sample. This value could be explained away in just a few words. • Lines 639 – 662: This text seems more pertinent to a geochemical methods paper and does not seem to be needed here. The discussion concerning the origin of Sr is interesting, but again, does not seem pertinent to the paper. • Overall, the authors should spend time editing and rewriting the sections dealing with Sr isotopes and the origin of Sr in the dolomite in order to make them shorter, but should still use the Sr isotope results in their paper (these results could be included in the text from lines 757-763). This section is interesting, but much of it seems tangential to the current paper, and should be removed and incorporated into a separate paper.

A second major criticism of the paper is the use of the the term "non-actualistic" when describing the conditions that led to the precipitation of the dolomites. "Non-actualistic" refers to periods when environmental conditions were so different from today that there is no modern analogue. For example, the occurrence of epeiric seas, or the resurgence of microbial carbonates following several mass extinctions. The conditions cited to have led to the growth of the dolomites only require minor modifications to modern models of dolomite precipitation (i.e., the occurrence of clay-rich aquitards preventing the input of meteoric waters), and so the authors should use different terminology here.

Overall, I recommend that the paper be accepted with major revisions, and that the editors work with the authors to cut down on the amount of text.

Specific comments:

Mud clasts vs. mudclasts: The authors use both spellings throughout the manuscript. The authors should separate the 2 words so that it reads "mud clasts".

Line 28: The use of the word non-actualistic is typically associated with unusual facies or intervals in Earth History. While the model proposed by the authors is certainly unusual, I would avoid using the term non-actualistic and instead perhaps state that there is no modern analogue for a similar system.

Line 43: Competing theories of what? I assume dolomite formation, but the authors need to be specific.

Lines 56-58: I would draw the attention of the authors to a recent paper published in Geology by Li et al. that documents the widespread precipitation of primary dolomite around the Permian – Triassic boundary.

Lines 61-62: What are the signatures indicative of a burial diagenetic overprint? The authors need to be specific.

Lines 68-70: This text is vague, and the authors need to explain what the dolomite phases are that are documented by Frisia and Wenk (1993) so the reader can better establish that these are burial diagenetic features.

Lines 88, 99 and 782: The authors use the term "Carnian platform", which is incorrect, as "Carnian" is a time term and a platform is a physical object. I would change the text to "Carnian-aged" and also add a modifier to state where the platform was. So, "Carnian-aged western Tethyan platform".

Lines 102-103: What is the evidence for seasonally wet conditions? The authors state in the abstract that the seasonally wet conditions make these dolomites special (non-actualistic in their terms), and so they need to provide evidence of the seasonally wet conditions.

Line 103: Which facies? Dolomite or clay? Or is this the entire sequence?

Line 104: Use of the term "extended" is confusing. Do the authors mean laterally-extensive? Extensive over time? Both?

Lines 104 and 107-109: Use of the term "a Germanic Keuper facies" is confusing. It's not clear to me if the authors are discussing a general facies type or a formational name, especially in lines 107-109 where they discuss paleogeographic separation between the Travenanzes Fm and the Germanic Keuper facies. Overall, the text in lines 107-109 is confusing and needs to be rewritten.

Line 111: One facies zone is the Germanic Keuper facies. What is the other facies zone? If it is the Travenanzes Fm, then the authors need to word this differently, since it is confusing to compare facies to formations. This can be solved by referring to "dolomitic facies of the Travenanzes Fm", for example.

Lines 114: I would replace "carbonate" with "dolomite" since the authors are attempting to prove that the dolomite is primary in origin, and use of the term "carbonate" here is not specific enough.

Lines 148-149: It's not clear to me why the authors interpret the Travenanzes Fm as having been deposited on a very flat surface based on the lithologies that make up the unit. The authors need to provide stronger evidence.

Line 190: Are there units that go with "...a spot size 5.0..."?

Lines 237-238: The authors need to provide units for their detection limits.

Line 292: Approximately how thick are the tempestite beds?

Line 293: I think the authors mean to state "megalodont teeth".

Lines 319-322: The ooids appear to have been micritized to me, based on the photo. Is this the case? If so, this needs to be mentioned in the description. If not, then the authors need a better photo.

Line 328: The authors need to be more specific in terms of what the measurements are measuring. Diameter? Thickness?

Line 329: "Pale" is not a color, it is a shade of color. The authors need to add a color after the word "pale".

Line 362: I'm not sure if the peloids are a rare type of allochem, or if subunits made up mostly of peloids are rare. Please clarify.

Line 436: I'm not sure what it meant by "It".

Lines 479-481: The authors state that the mud was unlithified, but also note the presence of rip-up clasts made of the same mud. The authors need to account for this difference, since rip-ups require at least semi-lithification to form.

Line 484: I'm not sure what the authors mean by "this type". Microfacies, perhaps?

Lines 491-492: The authors should include a reference at the end of this sentence.

Lines 492-493: This sentence seems out of place here since this is a discussion of processes within a possible ephemeral lake, and the previous text is trying to establish the larger depositional setting. In addition, I'm not sure that this text is necessary, since the mud is homogenous in composition, so stating that waves are responsible for homogenizing the mud is pure speculation without other evidence of wave action, like ripple marks.

Lines 511-513: I'm confused. Are the authors stating that the ooids are marine in origin, or lacustrine (like the ooids found at the Great Salt Lake). The authors need to be more clear as to what they believe the origin of the ooids are, and if lacustrine, provide modern examples, since ooids are rare in that setting.

Line 530: "in situ" should be in italics.

Lines 538-539: If the sediment is being plastically deformed, it must be at least partially lithified.

Lines 543-544: The authors need to cite a reference at the end of this sentence: "What is atypical for a modern sabkha is the large amount of detrital input."

Line 544: The authors need to cite a reference to support their contention that the Carnian was seasonally wet.

Line 554: The authors need to cite a reference that red color represents seasonally arid conditions in clays.

Line 556: I think the authors mean "after burial", not "after sedimentation".

Lines 679-680: This sentence is confusing, and needs to be rewritten. It could probably be shortened to just a few words and added to the end of the previous sentence.

Lines 691-696: These temperature ranges seem high, and therefore a reference to the temperature range of modern sabhkhas is needed. In addition, the authors also need to consider the effect of evaporation on oxygen isotope values and therefore temperature estimates from those values.

Lines 697-699: Why is this important?

Line 700: Why do the oxygen isotopes indicate a primary signature as opposed to a secondary signature?

Line 712-715: I'm not sure how these nodules relate to the cement rims surrounding the dolomicrite grains. I do agree with the formation mechanism for the nodules, but the authors need to add references to support their proposed formation mechanism.

Line 768: What kind of isotopes?

Figure 1:  c Change "positive areas" to "highlands" or "topographic highs".  c The authors need to define the following abbreviations in the figure caption: Drau., Mr, Wa, Kr, Be, Fr and Ly. • I don't see any Continental/Lacustrine areas on the map, but the symbol for this facies is provided on the key.

Figure 3: • 3A need to focus in on the homogenous dolomite bed, as it is currently difficult to see as the view is too far out to allow any details to be properly discerned. • The calcitic vertisol in 3b needs to be labelled. • Gypsum nodules and crack fills need to be more clearly labelled in 3d. • The view on 3f needs to be closer to allow the soft sediment deformation, and, in particular, the isoclinal fold, to be more clearly seen.

Figure 4: • The authors need to more clearly distinguish the mud clasts, as well as the coarser grained and finer grained layers in 4b. • The authors need to add arrows to 4d to point out the pseudomorphs. • The ooids in 4e appear to be micritized to me. This may be a reflection of the size/resolution of the photo. I would recommend that the authors show a close-up view of the ooids. • The feature labelled with a "P" in 4f is supposed to be a peloid, but it's not clear to me what the "P" is referring to on the photo. • The authors need to include boxes in 4g that shows the areas depicted in 4h and 4i. •

Figures 8 and 9: The captions need to be more detailed for all plots in both figures. What is the significance of each plot for the study?

Figure 10: What is the significance of the circled areas on the figure? This needs to be explained in the figure caption.

Figure 11: The authors need to note in the caption that Coorong Lagoon and Deep Springs Lake are modern dolomite deposits. Also, there is no mention of the Abu Dhabi sabkha in the caption, and it needs to be added to the caption.

Table 1: • I am unfamiliar with the term "laminate". The authors need to be more specific as to what this is. • This text at the bottom of the table is confusing and needs to be rewritten: "*Needs to be further subdivided into peloids, intraclasts, flat pebbles and clast of brittle deformation" I'm especially confused by the term "clast of brittle deformation". Table 2: I think the authors mean "Height" and not "Depth" as they refer to "Height" elsewhere.

Table 4: • Again, I think the authors mean "megalodont teeth". • It's not clear to me what the authors mean by "top", "bottom" or "part". •

Please also note the supplement to this comment:
https://www.solid-earth-discuss.net/se-2019-34/se-2019-34-RC2-supplement.pdf

[Figure]

**Supplement:**

[revised manuscript text omitted]

---

## Short Comment (SC1) · 14 Apr 2019

We are thankful to the Reviewer for his patience to look through our manuscript again. Moreover, we are thankful for now having the opportunity to respond directly to his comments.

With respect to Table 6, presenting the Sr isotope data, we understand that this is somewhat difficult to capture for the reader. The reason for the complexity is the fact that Sr-extraction procedures were intensively tested. Individual steps were adapted

along the way, based on the outcome of the previous step. We decided to show the full dataset for this submission, but we would be happy to provide a simplified table or plot that will be easier to grasp and which will directly correspond to the main text. The complete data table could be provided through an online repository. We we are happy to follow the instructions of the Editor.

Several other issues raised by the reviewer are related to the Sr-isotope analysis. To the comment that only a limited number of samples out of the 39 hand specimen collected in the field were analysed it is to say that several samples were not dolomite or a mixture of dolomite and clay. Upon petrographic inspection 11 samples were selected, which showed pure aphanotopic dolomite. It should be noted that the sole purpose of selecting three dolomite samples and two clay samples for elemental analysis was to test extraction efficiency. It was never intended to provide a full elemental analysis of dolomites through the section. An in-depth discussion in the sedimentological context would immediately raise the criticism that the sample selection was incomplete. We suggest to provide the data in Table 5 through an online repository. Furthermore, TOC and TIC measurements were performed on clay samples, not on the dolomites. As explained in line 208, the goal of these measurements was to select the clay samples with the lowest carbonate content, as a control. Also the data of Table 3 can be provided through an online repository. Since we have no TOC data from the dolomites, a further discussion of the organic role in dolomite formation, as suggested by the reviewer, would be rather speculative.

The reviewer further suggested that we discuss the microbial dolomite formation. This matter is currently rather controversially debated. Our manuscript does not provide much new insight on any microbial influence, nor is our interpretation affected by it. Therefore we prefer not to engage in an elongate discussion on this matter. We agree, however, that the microbial dolomite hypothesis should be briefly mentioned in the introduction and/or the discussion.

With the reviewer's conclusive statement that our study "provides an incremental step,

albeit a small one, in our general understanding of dolomite formation" we do not entirely agree. Our study provides more insight into the depositional environment and mechanism in an ancient system. Our work is, hence, of importance from a palaeo-environmental point of view, which should be valued for a geologically oriented journal as Solid Earth.

---

## Short Comment (SC2) · 14 Apr 2019

We are thankful to this reviewer for providing extensive comments throughout the manuscript. We agree with most suggestions and we will be happy to include them in a revision. They clearly help to improve the manuscript.

There are only a few points where we disagree or where we would be grateful for further clarification. Here we briefly discuss these points:

The reviewer finds the methodology description too extensive and suggests that this

part be significantly shortened. We agree so far that the description of TOC/TIC and elemental analyses, which are used to test the extraction method, could be reduced and included in the description of the Sr-isotope analysis. Also considerably shortening the manuscript could be achieved by exporting data tables to an online data repository. This would particularly concur with Reviewer Dr. Romanek who also commented that Table 6 is too complex. Table 6 could be provided as a simplified table or plot.

However, we disagree that the description of the Sr-extraction should be removed or referred to the literature. We would like to highlight that the extraction method is to a great part novel and designed for this particular study. It is crucial that contamination (e.g. by clay minerals) is exluded and to make sure the Sr-isotope values are truly measured from the dolomite phase. The precautions in the methodology are highly critical if we want to find a marine signal in dolomites embedded in large amounts of clay. Furthermore, we do not agree that the discussion of the origin of ionic solutions should be omitted or significantly shortened. The section on the origin of ionic solutions is very well embedded in the study as it leads up to the discussion that dolomite formed from seawater further below. This is the central part of this study as indicated already in the title. Removing this part would severely disrupt the context of the entire study. Furthermore, the Germanic Keuper was shown as a contrasting system, where dolomite forms in a similar setting but entirely disconnected from the sea. Therefore, this part should not be removed. To address the concerns of the reviewer, the authors are nevertheless prepared to go again through the manuscript to screen for possible parts that could be shortened, clarified or simplified.

Comment on homogeneous dolomite beds (Lines 492-493): Homogenization by wave actions is actually observed in many shallow water bodies of a few cm to dm depth. This process is very likely to homogenize the sediment, unlike in laminites showing separate clay and dolomite laminae. Clay fraction dolomite is transported in suspension and thus would not form wave ripples, unless the mud is clumped together as mud clasts.

The reviewer mentions twice that the ooids could have been micritized. However, it is

not clear to me how this could be shown, because ooids very often are already micritic. So how could we know if micrite is replaced by micrite?

In lines 511-513 we are essentially saying the same as the reviewer: Ooids may occur in both marine and lacustrine settings. In the present case they are rather marine because in the same bed Megalodon bivalves (not teeth) occur.

Comment to Line 538: On the contrary: lithified sediment cannot be plastically deformed. It would show brittle deformation.

Comment to Line 700: The oxygen isotopes indicate approximately modern sabkha temperatures, even taking into account the effect of evaporation. Therefore, this is not indicating overprint during burial diagenesis (see also Preto et al., 2015).

With all other comments we agree and we will be happy to follow the Reviewer's suggestions.

---

## Author Comment (AC1) · 25 Apr 2019

Response to Reviewer Dr. Romanek

We are thankful to the Reviewer for his patience to look through our manuscript again. Moreover, we are thankful for now having the opportunity to respond directly to his comments.

With respect to Table 6, presenting the Sr isotope data, we understand that this is somewhat difficult to capture for the reader. The reason for the complexity is the fact

that Sr-extraction procedures were intensively tested. Individual steps were adapted along the way, based on the outcome of the previous step. We decided to show the full dataset for this submission, but we would be happy to provide a simplified table or plot that will be easier to grasp and which will directly correspond to the main text. The complete data table could be provided through an online repository. We we are happy to follow the instructions of the Editor.

Several other issues raised by the reviewer are related to the Sr-isotope analysis. To the comment that only a limited number of samples out of the 39 hand specimen collected in the field were analysed it is to say that several samples were not dolomite or a mixture of dolomite and clay. Upon petrographic inspection 11 samples were selected, which showed pure aphanotopic dolomite. It should be noted that the sole purpose of selecting three dolomite samples and two clay samples for elemental analysis was to test extraction efficiency. It was never intended to provide a full elemental analysis of dolomites through the section. An in-depth discussion in the sedimentological context would immediately raise the criticism that the sample selection was incomplete. We suggest to provide the data in Table 5 through an online repository. Furthermore, TOC and TIC measurements were performed on clay samples, not on the dolomites. As explained in line 208, the goal of these measurements was to select the clay samples with the lowest carbonate content, as a control. Also the data of Table 3 can be provided through an online repository. Since we have no TOC data from the dolomites, a further discussion of the organic role in dolomite formation, as suggested by the reviewer, would be rather speculative.

The reviewer further suggested that we discuss the microbial dolomite formation. This matter is currently rather controversially debated. Our manuscript does not provide much new insight on any microbial influence, nor is our interpretation affected by it. Therefore we prefer not to engage in an elongate discussion on this matter. We agree, however, that the microbial dolomite hypothesis should be briefly mentioned in the introduction and/or the discussion.

With the reviewer's conclusive statement that our study "provides an incremental step, albeit a small one, in our general understanding of dolomite formation" we do not entirely agree. Our study provides more insight into the depositional environment and mechanism in an ancient system. Our work is, hence, of importance from a palaeo-environmental point of view, which should be valued for a geologically oriented journal as Solid Earth.

Response to anonymous Reviewer

We are thankful to this reviewer for providing extensive comments throughout the manuscript, especially also for correcting the language in the annotated manuscript. We agree with most suggestions and we will be happy to include them in a revision. They clearly help to improve the manuscript.

There are only a few points where we disagree or where we would be grateful for further clarification. Here we briefly discuss these points:

The reviewer finds the methodology description too extensive and suggests that this part be significantly shortened. We agree so far that the description of TOC/TIC and elemental analyses, which are used to test the extraction method, could be reduced and included in the description of the Sr-isotope analysis. Also considerably shortening the manuscript could be achieved by exporting data tables to an online data repository. This would particularly concur with Reviewer Dr. Romanek who also commented that Table 6 is too complex. Table 6 could be provided as a simplified table or plot.

However, we disagree that the description of the Sr-extraction should be removed or referred to the literature. We would like to highlight that the extraction method is to a great part novel and designed for this particular study. It is crucial that contamination (e.g. by clay minerals) is exluded and to make sure the Sr-isotope values are truly measured from the dolomite phase. The precautions in the methodology are highly critical if we want to find a marine signal in dolomites embedded in large amounts of clay. Furthermore, we do not agree that the discussion of the origin of ionic solutions

should be omitted or significantly shortened. The section on the origin of ionic solutions is very well embedded in the study as it leads up to the discussion that dolomite formed from seawater further below. This is the central part of this study as indicated already in the title. Removing this part would severely disrupt the context of the entire study. Furthermore, the Germanic Keuper was shown as a contrasting system, where dolomite forms in a similar setting but entirely disconnected from the sea. Therefore, this part should not be removed. To address the concerns of the reviewer, the authors are nevertheless prepared to go again through the manuscript to screen for possible parts that could be shortened, clarified or simplified.

Comment on homogeneous dolomite beds (Lines 492-493): Homogenization by wave actions is actually observed in many shallow water bodies of a few cm to dm depth. This process is very likely to homogenize the sediment, unlike in laminites showing separate clay and dolomite laminae. Clay fraction dolomite is transported in suspension and thus would not form wave ripples, unless the mud is clumped together as mud clasts.

The reviewer mentions twice that the ooids could have been micritized. However, it is not clear to me how this could be shown, because ooids very often are already micritic. So how could we know if micrite is replaced by micrite?

In lines 511-513 we are essentially saying the same as the reviewer: Ooids may occur in both marine and lacustrine settings. In the present case they are rather marine because in the same bed Megalodon bivalves (not teeth) occur.

Comment to Line 538: On the contrary: lithified sediment cannot be plastically deformed. It would show brittle deformation.

Comment to Line 700: The oxygen isotopes indicate approximately modern sabkha temperatures, even taking into account the effect of evaporation. Therefore, this is not indicating overprint during burial diagenesis (see also Preto et al., 2015).

With all other comments we agree and we will be happy to follow the Reviewer's suggestions.

---

## Author Response (AR1)

Dear Editor,

Please find enclosed our revised manuscript "Precipitation of dolomite from seawater on a Carnian coastal plain (Dolomites, northern Italy): evidence from carbonate petrography and Sr-isotopes" for your consideration as publication in *Solid Earth*.

The reviewer's suggestions allowed us to improve the manuscript significantly and we are thankful for their efforts. We have already responded in a general way as part of the open discussion. Here we provide specific responses to each comment. The indicated line numbers refer to the clean version of the manuscript.

All data from the former Tables 3-6 were uploaded to the Pangaea online repository under the access number PDI-20535. The data report awaits confirmation by the administrator.

We hope you will find the responses and the changes in the annotated manuscript satisfactory.

Sincerely yours,

Patrick Meister

**Referee #1 (Chris Romanek)**

This manuscript was very well written and it was a pleasure to read. In fact, there were very few places where improvements could be suggested. The summary information provided in the introduction was appropriate and insightful, and the methodology and analytical procedures were explained in a straightforward way. The investigators presented convincing petrographic and geochemical evidence that supports their interpretation for the depositional environment of the Travenanzes dolomites. The Sr isotope data are remarkably consistent with Triassic seawater throughout the length of the section and they only show hints of a continental signature with the most aggressive leaching procedures. The stable O-isotope data are consistent with a marine signature and the C-isotopes demonstrate the incorporation of oxidized organic matter in texturally distinct samples. Overall, the data appear to be straightforward and easy to interpret.

We are thankful to the Reviewer for his patience to look through our manuscript again, and we are pleased about the encouraging assessment given here. Thereby we would like to highlight that Reviewer Romanek states that "the methods are explained in a straightforward way" and he does in no way suggest that the manuscript is too long or that the methods should be cut down. This seems therefore to be a particular opinion of the anonymous Reviewer 2.

The sequential extraction work (e.g., Table 6) for the Sr-isotope work could be presented better so the reader can understand why various procedures and reagents (e.g., NaCl, AcOH, HCl) were being used. The TIC/TOC results could be integrated in more substantial ways, e.g., perhaps TOC could be related to the development of dolomite nodules during microbial sulfate reduction."

In this point, we agree with both reviewers. Table 6 has now been uploaded to the Pangaea data repository, and we provide a better overview of the leaching procedure by showing the results graphically, as a new Figure 10.

It should be noted that only clay samples were analysed for TIC and TOC, for the purpose to select the sample with the lowest carbonate content. Also these data are now available from the Pangaea repository.

Several aspects of the present manuscript, that were identified in my previous review, still remain: 1) a general lack of engagement with the bulk elemental data (i.e., Table 5), 2) although 39 samples were collected in this study, it appears only a handful of these are presented in the manuscript for analysis, and 3) the manuscript does not substantially engage the potential for microbial origins although the subject is broached in general ways.

1.) It should be noted that the sole purpose of selecting three dolomite samples and two clay samples for elemental analysis was to test extraction efficiency. It was never intended to provide a full elemental analysis of dolomites through the section. An in-depth discussion in the sedimentological context would immediately raise the criticism that the sample selection was incomplete. Table 5 is also in the repository.

2.) Several samples of the 39 hand specimens collected in the field were claystone or a mixture of dolomite and clay. Upon petrographic inspection 11 samples were micro-drilled to analyse the most pure aphanotopic dolomite.

3.) The reviewer suggests that we discuss the microbial dolomite formation. This matter is currently rather controversially debated. Our manuscript does not provide much new insight on any microbial influence, nor is our interpretation affected by it. Therefore we prefer not to engage in an elongate discussion on this matter. We agree, however, that the microbial dolomite hypothesis should be briefly mentioned. A short section was added in the discussion (line 720).

Nevertheless, and as stated in my previous review, ". . . my overall impression is that this manuscript provides a plausible interpretation for the origin of Travenanzes dolomites. This contribution provides an incremental step, albeit a small one, in our general understanding of dolomite formation and more specifically dolomite formation along the Tethyan margin and I feel it is acceptable for publication ...", although slight modifications are warranted that can be supervised by the associate editor.

With the reviewer's conclusive statement that our study "provides an incremental step, albeit a small one, in our general understanding of dolomite formation" we do not entirely agree. Our study provides more insight into the depositional environment and mechanism in an ancient system. Our work is, hence, of importance from a palaeo-environmental point of view, which should be valued for a geologically oriented journal as *Solid Earth*.

**Anonymous Referee #2**

I have read through the paper by Reider and others entitled: "Precipitation of dolomite from seawater on a Carnian coastal 1 plain (Dolomites, northern Italy): evidence from carbonate petrography and Sr-isotopes". I find the paper to be an interesting contribution to our understanding of the processes that led to the formation of primary dolomite in the sedimentary rock record and the authors use some innovative methods to prove a primary origin for the dolomite.

We are thankful to this reviewer for providing extensive comments throughout the manuscript, and for providing annotations in the manuscript. We agree with most suggestions, which we included in the revision. Below we explain how each point was addressed, or, in a few cases, why we disagree with the Reviewer's comment.

However, my main criticism about the paper is that it is too long in its current state, and should be shortened. Examples of text that needs to be edited or cut out altogether include:

- The authors spend approximately 4 pages describing their methods, which could be cut down to at least half that length by referring to similar methods in other papers and describing their methods in less detail.

    We found a way to shorten the section by including the description of elemental and TOC/TIC analyses as part of the Sr-isotope analytics. Elemental and TOC analyses were only performed to test extraction efficiency of the sequential extraction for Sr-isotopes. Also the results section was considerably shortened and tables were moved to the Pangaea data repository.

- Related to the previous example, the paper contains data (e.g., %TOC, some of the elemental data, etc.) that seems unnecessary to the overall study. I would recommend that the authors carefully go through their manuscript and remove any data that is not deemed essential to their manuscript.

    Done. TOC data were only measured in the clay minerals and are not discussed in the discussion. Both TOC and elemental data are now deposited in the Pangaea repository.

- The inclusion of 6 tables seems excessive. The number of tables should be reduced to one or two, with the extra tables relegated to a "Supplemental Materials" section.

    Done. Table 1 is provided as supplemental material. Table 2 is shown as an inset in Fig. 7. Tables 3 through 6 are now deposited in the Pangaea data repository.

- It is not clear to me why the authors include analyses of the Germanic Keuper dolomites (lines 359-371) in this paper.

    The sentence in line 185 in the introduction was rephrased to clarify the purpose of analysing Keuper dolomites: "To demonstrate contrasting origin of ionic solutions, Sr-isotope values were compared to values from dolomites from the Germanic Keuper, that are of clear continental origin, and to values in modern dolomites showing marine and/or continental influence."

- I am not an expert on Sr geochemistry, but it's not clear to me why the authors spend so much time discussing sequential extractions (lines 425-472). It seems to me that this text could be reduced.

    The sequential extraction is absolutely critical to provide Sr-isotope data from pure uncontaminated dolomite, in particular if they are embedded in so much clay. The authors do not know of any other study, where the extraction procedure was so rigorously tested. This method must be described in detail here and cannot be shortened. In accordance with Reviewer 1, we provide a graphical representation of the leaching results, which should provide more clarity to the reader.

- Lines 622-638 also seems unnecessary to the paper, as the authors explain one anomalous value from one sample. This value could be explained away in just a few words.

    The first sentence of the section was rephrased (line 619). This discussion is not only about one outlier but about the influences that can provide more radiogenic values in general. This section should not be removed.

- Lines 639 – 662: This text seems more pertinent to a geochemical methods paper and does not seem to be needed here. The discussion concerning the origin of Sr is interesting, but again, does not seem pertinent to the paper.

We refer the Reviewer to the goal of the study in the introduction: "… to determine if ionic solutions conducive to dolomite formation were derived from seawater or from continental runoff."

Hence, the discussion of the origin of Sr is central to this study and cannot be removed.

- Overall, the authors should spend time editing and rewriting the sections dealing with Sr isotopes and the origin of Sr in the dolomite in order to make them shorter, but should still use the Sr isotope results in their paper (these results could be included in the text from lines 757-763). This section is interesting, but much of it seems tangential to the current paper, and should be removed and incorporated into a separate paper.

See comment above.

A second major criticism of the paper is the use of the the term "non-actualistic" when describing the conditions that led to the precipitation of the dolomites. "Non-actualistic" refers to periods when environmental conditions were so different from today that there is no modern analogue. For example, the occurrence of epeiric seas, or the resurgence of microbial carbonates following several mass extinctions. The conditions cited to have led to the growth of the dolomites only require minor modifications to modern models of dolomite precipitation (i.e., the occurrence of clay-rich aquitards preventing the input of meteoric waters), and so the authors should use different terminology here.

We believe the term "non-actualistic" is still correct. First, thick successions of fine-grained, fabric-preserving dolomite as the Triassic Travenanzes and Dolomia Principale do not form today – it is indeed an unusual facies. Second, "non-actualistic" (also: "non-uniformitarian") not only implies extreme events rare in Earth history but more generally should include situations where processes were similar, but boundary conditions were different from today. Accordingly, a process may have been different in duration, scale, or rate from any modern analogue. Dolomite formation is a very good example where delicate changes in boundary conditions could have made the difference. There are many aspects about the Triassic Tethys that imply non-actualism, such as a different sealevel and area of epeiric platforms, Tethys seawater chemistry, atmospheric $pCO_2$, climate, all of which may have influenced the formation of dolomite. Hence, the present as the key to the past only partially works under these circumstances and must be used with caution.

Overall, I recommend that the paper be accepted with major revisions, and that the editors work with the authors to cut down on the amount of text.

We tried our best to improve the manuscript, and we incorporated most suggestions by the reviewer, except for shortening the discussion on the origin of ionic solutions, which is central to this study.

**Specific comments:**

Mud clasts vs. mudclasts: The authors use both spellings throughout the manuscript. The authors should separate the 2 words so that it reads "mud clasts".

Done

Line 28: The use of the word non-actualistic is typically associated with unusual facies or intervals in Earth History. While the model proposed by the authors is certainly unusual, I would avoid using the term non-actualistic and instead perhaps state that there is no modern analogue for a similar system.

See discussion above.

 Competing theories of what? I assume dolomite formation, but the authors need to be specific.

It should say "dolomite formation" (was added to the sentence).

Lines 56-58: I would draw the attention of the authors to a recent paper published in Geology by Li et al. that documents the widespread precipitation of primary dolomite around the Permian – Triassic boundary.

We are thankful for this reference, which was included.

Lines 61-62: What are the signatures indicative of a burial diagenetic overprint? The authors need to be specific.

Rephrased to: "… show oxygen isotope signatures of diagenetic overprint at burial temperature"

Lines 68-70: This text is vague, and the authors need to explain what the dolomite phases are that are documented by Frisia and Wenk (1993) so the reader can better establish that these are burial diagenetic features.

Further explanation is provided with reference to the recent publication (Meister and Frisia, 2019). See line 70.

Lines 88, 99 and 782: The authors use the term "Carnian platform", which is incorrect, as "Carnian" is a time term and a platform is a physical object. I would change the text to "Carnian-aged" and also add a modifier to state where the platform was. So, "Carnian-aged western Tethyan platform".

Done

Lines 102-103: What is the evidence for seasonally wet conditions? The authors state in the abstract that the seasonally wet conditions make these dolomites special (nonactualistic in their terms), and so they need to provide evidence of the seasonally wet conditions.

The sentence was reorganized to clarify that the large amount of distal riverine silicilastic input, and the presence of vertic paleosols suggesting vertical movements of the water table implies at least seasonally humid conditions. See lines 105-107.

Line 103: Which facies? Dolomite or clay? Or is this the entire sequence?
It should say "facies association".

Line 104: Use of the term "extended" is confusing. Do the authors mean laterally extensive? Extensive over time? Both?
Yes it should be "extended in space". The sentence was re-organized to clarify this point.

Lines 104 and 107-109: Use of the term "a Germanic Keuper facies" is confusing. It's not clear to me if the authors are discussing a general facies type or a formational name, especially in lines 107-109 where they discuss paleogeographic separation between the Travenanzes Fm and the Germanic Keuper facies. Overall, the text in lines 107-109 is confusing and needs to be rewritten.

The reviewer is right that the Germanic Keuper is a palaeogeographic region and not a facies. We are trying to distinguish between the palaeogeographic regions of the Germanic Basin and the Alpine Tethys region. However, the continental facies association found in Germanic appears to reach far beyond just the Germanic Basin. We can see red clays with intercalated dolomites in both an endorheic setting (playa lake – alluvial plain facies association) or in a setting linked to the Tethyan sea (coastal lake – dirty sabkha facies association). The lithological evidence is scarce (except in the few beds showing marine fossils). Sr isotopes can help us to distinguish the two facies associations.

Line 111: One facies zone is the Germanic Keuper facies. What is the other facies zone? If it is the Travenanzes Fm, then the authors need to word this differently, since it is confusing to compare facies to formations. This can be solved by referring to "dolomitic facies of the Travenanzes Fm", for example.
Yes, we agree. See comment above.

Lines 114: I would replace "carbonate" with "dolomite" since the authors are attempting to prove that the dolomite is primary in origin, and use of the term "carbonate" here is not specific enough.
Done (just to be cautious, we add in brackets "or a precursor carbonate phase")

Lines 148-149: It's not clear to me why the authors interpret the Travenanzes Fm as having been deposited on a very flat surface based on the lithologies that make up the unit. The authors need to provide stronger evidence.
This becomes obvious from the stratigraphic context. Gattolin et al. (2013/15) show very clearly the stratigraphic relationships where the deep basins are filled up and sealed by the laterally extensive clay depostits of the Travenanzes Fm. The palaeogeography has been established by Breda and Preto (2011) showing the interfingering with alluvial fans and marine carbonates over tens of kilometres. The sentence was rephrased (line 157).

Line 190: Are there units that go with ". . .a spot size 5.0. . ."?
There is no unit for the spot size. "5" is just the number of the spot size chosen.

Lines 237-238: The authors need to provide units for their detection limits.
The unit is µmol/g for all measurements, as indicated in the same sentence (line 260).

Line 292: Approximately how thick are the tempestite beds?
Ca. 20-cm-thick.

Line 293: I think the authors mean to state "megalodont teeth".
It should say "megalodont bivalves"

Lines 319-322: The ooids appear to have been micritized to me, based on the photo. Is this the case? If so, this needs to be mentioned in the description. If not, then the authors need a better photo.
The ooids show concentric, micritic layers. I think it is not possible to say whether they were micritized or originally micritic. In fact, ooids are often micritic.

Line 328: The authors need to be more specific in terms of what the measurements are measuring. Diameter? Thickness?
It should be "diameter"

Line 329: "Pale" is not a color, it is a shade of color. The authors need to add a color after the word "pale".
It should say "pale grey"

Line 362: I'm not sure if the peloids are a rare type of allochem, or if subunits made up mostly of peloids are rare. Please clarify.
It should say that there are only a few peloids in the thin section. Both oolites and peloidal grainstone occur as part of a distal shoal facies in the Weser Fm. (Seegis). See line 363.

Line 436: I'm not sure what it meant by "It".
The pure celestine. The whole section was reorganized.

Lines 479-481: The authors state that the mud was unlithified, but also note the presence of rip-up clasts made of the same mud. The authors need to account for this difference, since rip-ups require at least semi-lithification to form.
The rip-up clasts often show ductile deformation and are well rounded. They were clearly unlithified. Most likely the cohesive mud was sticking together (probably with a consistency like cottage cheese). Some clasts show brittle deformation: those ones may be semi-lithified.

Line 484: I'm not sure what the authors mean by "this type". Microfacies, perhaps?
Yes, it should be the homogeneous aphanotopic dolomite described here, we explicited it.

Lines 491-492: The authors should include a reference at the end of this sentence.
This was suggested by Breda and Preto (2011). The reference was added.

Lines 492-493: This sentence seems out of place here since this is a discussion of processes within a possible ephemeral lake, and the previous text is trying to establish the larger depositional setting. In addition, I'm not sure that this text is necessary, since the mud is homogenous in composition, so stating that waves are responsible for homogenizing the mud is pure speculation without other evidence of wave action, like ripple marks.
The sentence explains why the sediment is homogenous, as opposed to laminated with separate clay and carbonate mud couplets observed in the laminated facies. The homogenization is explained by mixing upon wave action, which is commonly observed in shallow ephemeral lakes (e.g. Deep Springs Lake, Coorong Lakes, Lake Neusiedl, etc.). No bed forms, such as ripple marks, are formed because the dolomite is in the clay size fraction and transported in suspension. To better embed the sentence in the context of the section, the following sentence was added: "…, which is often observed in ephemeral lake settings, explaining the formation of homogeneous dolomite beds."

Lines 511-513: I'm confused. Are the authors stating that the ooids are marine in origin, or lacustrine (like the ooids found at the Great Salt Lake). The authors need to be more clear as to what they believe the origin of the ooids are, and if lacustrine, provide modern examples, since ooids are rare in that setting.
Marine fossils that occur in the same bed point to a marine origin in this case. The sentence is not important and was removed.

Line 530: "in situ" should be in italics.
Done

Lines 538-539: If the sediment is being plastically deformed, it must be at least partially lithified.
Lithified means that it is actually cemented by a mineral phase. If this is the case, the sediment can only break discretely and can no longer show plastic deformation. As we see plastic deformation, the sediment must be unlithified but cohesive.

 The authors need to cite a reference at the end of this sentence: "What is atypical for a modern sabkha is the large amount of detrital input."

"Detrital input" is perhaps misleading because actual sabkhas may receive Aeolian input. What we mean is the large amounts of clay, which is derived from river flooding (Breda and Preto, 2011), which requires at least episodically humid conditions. This matter is already discussed in the introduction and is again addressed later in the discussion. For this reason we shall not engage further in this matter here. "detrital" was changed to "clay", and in brackets we add "(see discussion below)".

 The authors need to cite a reference to support their contention that the Carnian was seasonally wet.

Perhaps not the Carnian overall, but the depositional environment experienced episodic, most probably seasonal, fluvial input. This is well established by the regional facies reconstruction by Breda and Preto (2011), and probably represents the tail (the last pulses) of the Carnian Pluvial Episode.

 The authors need to cite a reference that red color represents seasonally arid conditions in clays.

A reference to Sheldon (2005), and a discussion as to why drainage was reduced, was added in line 547.

 I think the authors mean "after burial", not "after sedimentation".

Sentence rephrased to: "… internal brecciation, which must have occurred after sedimentation". Then in the next sentence we say that internation brecciation is also typical in calcretes, hence not "after burial".

 This sentence is confusing, and needs to be rewritten. It could probably be shortened to just a few words and added to the end of the previous sentence.

Sentence rephrased.

 These temperature ranges seem high, and therefore a reference to the temperature range of modern sabhkhas is needed. In addition, the authors also need to consider the effect of evaporation on oxygen isotope values and therefore temperature estimates from those values.

A reference to Hsü and Schneider (1973) was added.

The effect of evaporation on the oxygen isotope values of the water is already taken into account (see line 692) with reference to McKenzie et al. (1980) and McKenzie (1981).

 Why is this important?

The trend in $d^{18}O$ is observed. We only provide possible explanations here. It is a matter of ongoing investigation; we cannot say more at the moment.

 Why do the oxygen isotopes indicate a primary signature as opposed to a secondary signature?

The reviewer is correct that this is somewhat overstated. While the matter of temperature is being further investigated, we moderate the wording to: "… the oxygen isotope data do not imply any post-depositional overprint."

Line 712-715: I'm not sure how these nodules relate to the cement rims surrounding the dolomicrite grains. I do agree with the formation mechanism for the nodules, but the authors need to add references to support their proposed formation mechanism.
This is just a mass balance. If unlithified mud becomes lithified after sedimentation or during burial, the aphanotopic cement filling the interstitial space between the micro-scale crystals incorporates the isotopic composition of the surrounding interstitial fluid. The dissolved inorganic carbon most likely carries a more negative $d^{13}C$ value, due to decomposition of organic matter. As said, this is a simple consequence of mass balance.

Line 768: What kind of isotopes?
Sr-isotope values.

Figure 1: âAc Change "positive areas" to "highlands" or "topographic highs". âAc The authors need to define the following abbreviations in the figure caption: Drau., Mr, Wa, Kr, Be, Fr and Ly. âAc I don't see any Continental/Lacustrine areas on the map, but the symbol for this facies is provided on the key.
Done.

Figure 3: âAc 3A need to focus in on the homogenous dolomite bed, as it is currently difficult to see as the view is too far out to allow any details to be properly discerned. âAc The calcitic vertisol in 3b needs to be labelled. âAc Gypsum nodules and crack fills need to be more clearly labelled in 3d. âAc The view on 3f needs to be closer to allow the soft sediment deformation, and, in particular, the isoclinal fold, to be more clearly seen.
Fig. 3A shows the large-scale bedding relationship of homogeneous dolomite beds. The image size was further increased. For the description of the aphanitic microstructure we refer to the subsequent section. In 3b, the vertisol was graphically indicated. Gypsum in 3d was labelled with "Gy". Fig. 3f was zoomed in to better show the isoclinal fold.

Figure 4: âAc The authors need to more clearly distinguish the mud clasts, as well as the coarser grained and finer grained layers in 4b. âAc The authors need to add arrows to 4d to point out the pseudomorphs. âAc The ooids in 4e appear to be micritized to me. This may be a reflection of the size/resolution of the photo. I would recommend that the authors show a close-up view of the ooids. âAc The feature labelled with a "P" in 4f is supposed to be a peloid, but it's not clear to me what the "P" is referring to on the photo. âAc The authors need to include boxes in 4g that shows the areas depicted in 4h and 4i. âAc
Done. Fig. 4d is full of pseudomorphs, so arrows are only pointing out examples.
Ooids are micritic. This is already explained in the text "… consist of microcrystalline dolomite and lack a radial structure". They most likely are recrystallized because dolomitic ooids are, not to our knowledge, observed in modern environments. The replacement must have been mimetic, i.e. replicating the micritic structure down to the micron size. This needs further examination by SEM in future studies.

Figures 8 and 9: The captions need to be more detailed for all plots in both figures. What is the significance of each plot for the study?
Further explanations were added.

Figure 10: What is the significance of the circled areas on the figure? This needs to be explained in the figure caption.
Circled datapoints are clay samples or samples of nodules containing clay. Information was added in the figure caption.

Figure 11: The authors need to note in the caption that Coorong Lagoon and Deep Springs Lake are modern dolomite deposits. Also, there is no mention of the Abu Dhabi sabkha in the caption, and it needs to be added to the caption.
Figure caption was re-organized accordingly.

Table 1: âAc I am unfamiliar with the term "laminate". The authors need to be more specific as to what this is. âAc This text at the bottom of the table is confusing and needs to be rewritten: "*Needs to be further subdivided into peloids, intraclasts, flat pebbles and clast of brittle deformation" I'm especially confused by the term "clast of brittle deformation".
It should say "laminite" (not laminate). This refers to the classical "Loferites" or "algal laminites", except in this case they may not be algal (or microbial). The table was changed to a neutral terminology: "laminated dolomite". The Table will be provided as online supplemental material (Table S1).

Table 2: I think the authors mean "Height" and not "Depth" as they refer to "Height" elsewhere.
Yes, "height" is correct. The table is now incorporated as inset in Fig. 7.

Table 4: âAc Again, I think the authors mean "megalodont teeth". âAc It's not clear to me what the authors mean by "top", "bottom" or "part". âAc
It should say "*Megalodon* bivalves". "top" and "bottom" refers to the position within the thin section. This is of no meaning for the interpretation and was therefore removed. The data are now available from the Pangaea repository.

Please also note the supplement to this comment: https://www.solid-earth-discuss.net/se-2019-34/se-2019-34-RC2-supplement.pdf    Interactive comment on Solid Earth Discuss. https://doi.org/10.5194/se-2019-34, 2019.
We are extremely thankful to Reviewer 2 for very nicely revising grammar and style of our manuscript.

[revised manuscript text omitted]

Patrick 23-4-2019 16:52

Patrick 23-4-2019 16:52

Patrick 25-4-2019 23:20

Patrick 13-5-2019 01:52

Patrick 13-5-2019 01:52

Patrick 13-5-2019 01:52

Patrick 23-4-2019 16:53

Patrick 23-4-2019 16:54

Patrick 23-4-2019 16:54

Patrick 23-4-2019 16:54

Patrick 23-4-2019 16:54

Patrick 28-4-2019 18:12

Patrick 23-4-2019 16:50

Patrick 23-4-2019 16:50

Patrick 23-4-2019 16:51

Patrick 23-4-2019 16:50

Patrick 23-4-2019 18:15

section show sub-millimetre lamination that is only visible under the microscope, where it consists of alternating layers of light (locally coarser) and dark aphanotopic dolomite.

The clay content in the homogeneous beds is generally low. A few beds (e.g. at 33.5 m in the section) consist of silty or sandy dolomite, as reflected in a high abundance of detrital quartz in thin section. Pseudomorphs after gypsum occur in a dolomite bed at 120 m (Fig. 4c, d). Moldic porosity occurs within aphanotopic dolomite layers at 43, 65 and 89 m. These correspond to the tempestite beds observed in outcrop (cf. Breda and Preto, 2011).

One dolomite bed, located at 64 m in the section, appears homogeneous at outcrop scale, but consists of oolitic grainstone and lacks both an aphanotopic and a cement matrix (Fig. 4e).

Ooids show concentric, micritic layers and are either hollow (where the cores may have been dissolved) or filled with sparite, and are surrounded with an isopachous cement rim.

*Laminated dolomites*

Laminated dolomites occur in the upper part of the clay-rich interval, between 90 and 110

m in the section (Fig. 4f-i). In the field, the laminated dolomites show an alternation between light grey dolomite laminae and dark grey to black clay laminae. Some dolomite laminae are bent upward and are reminiscent of pseudo-teepee structures (Fig. 4f); the space within the teepee is sometimes infilled with sparry cement. In addition, the bending of the laminae towards the upward directed cuspids is reminiscent of load structures (dish structures), but they also may represent desiccation cracks. The laminae are frequently ripped apart and fragments of laminae occur reworked as flat pebbles embedded in an aphanotopic dolomite matrix (Fig. 4g). Some laminae show a microsparitic appearance and laminar fenestral porosity. In some laminae a clotted peloidal fabric is observed (e.g in Fig. 4f). Laminae are typically graded, whereby the upper part is darker, indicating an increase in the clay content (Fig. 4h, i). The top of the laminae is often truncated by an erosion surface, and rip-up clasts of the fine mud are embedded in the overlying coarse layer. Some laminated dolomites

Patrick 23-4-2019 16:49

Patrick 23-4-2019 16:49

Patrick 23-4-2019 16:48

Patrick 23-4-2019 16:48

Patrick 23-4-2019 16:47

Patrick 23-4-2019 16:48

Patrick 14-5-2019 12:27

Patrick 14-5-2019 12:25

Patrick 14-5-2019 12:26

Patrick 23-4-2019 16:47

Patrick 13-5-2019 01:58

Patrick 13-5-2019 01:58

Patrick 13-5-2019 01:57

Patrick 14-5-2019 13:56

Patrick 13-5-2019 02:08

Patrick 23-4-2019 16:44

Patrick 23-4-2019 16:44

Patrick 23-4-2019 16:44

Patrick 23-4-2019 16:45

Patrick 23-4-2019 16:45

Patrick 23-4-2019 16:45

Patrick 23-4-2019 16:45

Patrick 23-4-2019 16:45

Patrick 23-4-2019 16:43

Patrick 23-4-2019 16:43

[revised manuscript text omitted]

---

## Editor Decision (ED1)

[revised manuscript text omitted]

Figure 1

[Figure]

Figure 2

[Figure]

Figure 3

[Figure]

Figure 4

[Figure]

Figure 4 continued

[Figure]

Figure 5

[Figure]

Figure 6

[Figure]

Figure 7

[Figure]

Figure 8

[Figure]

Figure 9

[Figure]

Figure 10

[Figure]

Figure 11

[Figure]

Figure 12